# Mitochondrial uncoupling reveals a novel therapeutic opportunity for p53-defective cancers

R. Kumar [1,2], L. Coronel[1,3], B. Somalanka[3], A. Raju[4], O.A. Aning[4], O. An[5], Y.S. Ho[6], S. Chen[6], S.Y. Mak[6], P.Y. Hor[3], H. Yang[5], M. Lakshmanan[4,7], H. Itoh[8], S.Y. Tan[4,7], Y.K. Lim[9], A.P.C. Wong[10], S.H. Chew[10], T.H. Huynh[11,12], B.C. Goh[5,12,13], C.Y. Lim[8,14], V. Tergaonkar[4,14] & C.F. Cheok[1,4,7,14]

There are considerable challenges in directly targeting the mutant p53 protein, given the large heterogeneity of p53 mutations in the clinic. An alternative approach is to exploit the altered fitness of cells imposed by loss-of-wild-type p53. Here we identify niclosamide through a HTS screen for compounds selectively killing p53-deficient cells. Niclosamide impairs the growth of p53-deficient cells and of p53 mutant patient-derived ovarian xenografts. Metabolome profiling reveals that niclosamide induces mitochondrial uncoupling, which renders mutant p53 cells susceptible to mitochondrial-dependent apoptosis through preferential accumulation of arachidonic acid (AA), and represents a first-in-class inhibitor of p53 mutant tumors. Wild-type p53 evades the cytotoxicity by promoting the transcriptional induction of two key lipid oxygenation genes, *ALOX5* and *ALOX12B*, which catalyzes the dioxygenation and breakdown of AA. Therefore, we propose a new paradigm for targeting cancers defective in the p53 pathway, by exploiting their vulnerability to niclosamide-induced mitochondrial uncoupling.

[1] IFOM-p53Lab Joint Research Laboratory, IFOM, Milan 20139, Italy. [2] Duke–NUS Graduate Medical School, 8 College Road, Singapore 169857, Singapore. [3] p53 Laboratory, Agency for Science Technology and Research, Singapore 138648, Singapore. [4] Institute of Molecular and Cell Biology, Agency for Science Technology and Research, Singapore 138673, Singapore. [5] Cancer Science Institute of Singapore, National University of Singapore, Singapore 117599, Singapore. [6] Bioprocessing Technology Institute, Agency for Science Technology and Research, Singapore 138668, Singapore. [7] Department of Pathology, Yong Loo Lin School of Medicine, National University of Singapore, Singapore 119074, Singapore. [8] Skin Research Institute of Singapore, Agency for Science Technology and Research, Singapore 138648, Singapore. [9] Department of Gynaecological Oncology, KK Women's and Children's Hospital, Singapore 229899, Singapore. [10] Department of Pathology and Laboratory Medicine, KK Women's and Children's Hospital, Singapore 229899, Singapore. [11] Laboratory of Molecular Endocrinology, National Cancer Centre Singapore, Singapore 169610, Singapore. [12] Department of Pharmacology, Yong Loo Lin School of Medicine, National University of Singapore, Singapore 117600, Singapore. [13] National University Cancer Institute, Singapore 119074, Singapore. [14] Department of Biochemistry, Yong Loo Lin School of Medicine, National University of Singapore, Singapore 117596, Singapore. These authors contributed equally: R. Kumar, L. Coronel. Correspondence and requests for materials should be addressed to C.F.C. (email: cheokcf@imcb.a-star.edu.sg)

Mutation in p53 is the one of the most common occurrences across various types of human cancers[1,2]. Targeting p53 deficiency therefore holds great promise for a large cohort of human cancers. In particular, targeting vulnerability created by the loss-of-wild-type p53 functions may provide a means to target a broad spectrum of p53 mutations, including hotspot missense and non-sense mutations. It is hoped that such an approach widens the therapeutic window of the drug, leaving normal cells minimally affected.

Diverse mechanisms induce p53 activation, which in turn affects multiple processes in metabolism, DNA repair, and cell cycle control[3,4]. The cellular context and extent of p53 stimulation determine whether p53 activation confers protection and cellular homeostasis, or induces apoptosis[5]. The subtle response of p53 to transient metabolic stress is needed for cells to adapt and survive and may even provide unexpected anti-tumorigenic functions[6,7]. Unraveling other protective functions of p53 that may be exploited in a chemotherapy context, i.e., in protecting normal cells while allowing mutant cells to be fully exposed to the cytotoxicity, is beneficial for designing clinical strategies against p53 mutant tumors.

Since p53 governs a wide array of networks, deletion of p53 functions is likely associated with an altered "fitness" in cells that might undermine their genome stability and metabolic adaptability. Evidence that deletion of p53 predisposes cells to serine starvation and ROS-induced death from the absence of PI5P4Kα/β exemplifies the potential use of synthetic lethality in eliminating p53 mutant cells[8,9]. However, despite the advancements in the discovery of new functions of p53 in diverse metabolic pathways, pharmacologic manipulation of these complex pathways for therapeutic benefit still remains challenging. Drug strategies that can trigger optimal protective p53 activation while inducing cytotoxicity in mutant cells will be ideal for therapeutic translation.

Niclosamide is an oral salicylanilide derivative approved by the US Food and Drug Administration (FDA) since 1960 for human use in the treatment of intestinal tapeworm infections[10,11]. Niclosamide is a hydrogen ionophore that translocates protons across the mitochondrial membrane resulting in mitochondrial uncoupling[12] and futile cycles of glucose and fatty acid oxidation[12–14]. Its action as a mild mitochondrial uncoupler is sufficient to kill tapeworms residing in the gastrointestinal tract. It has an excellent safety profile in human (unlike 2,4 dinitrophenol DNP)[15,16] as transient mild mitochondrial uncoupling is tolerable in normal cells[17–19]. Niclosamide is recently characterized for uses in diabetes[12], Zika virus infection[20] and various cancer indications, including human glioblastoma tumors[21], colon and ovarian cancer cells in vitro[22,23]. Reports suggest that niclosamide inhibits tumor growth promoting pathways, including WNT/beta-catenin, STAT3, Notch, and mTOR pathways, however, its exact antitumor mechanism is not entirely clear[24–29].

In this study, we designed an image-based co-culture screen to identify compounds synthetically lethal to p53-deficient tumor cells. We identified the use of niclosamide in simultaneously activating a p53 prosurvival function in wild-type cells while effectively promoting apoptosis in cells lacking p53 function, of both human and mouse origins. Mechanistically, p53 activation is coupled to the transcriptional induction of lipid oxygenation genes that counteracts the metabolic stress conferred by niclosamide. Together, our study reveals an unexpected metabolic vulnerability in p53-defective cells that is targeted by niclosamide and supports the potential use of niclosamide as a first-in-class drug against a broad spectrum of tumors deficient in p53 functions.

## Results

**Repurposing niclosamide for targeting p53-deficiency**. We performed HTS screening of 1600 FDA approved compounds from the PHARMAKON Library. To select for compounds that synthetically kill p53-deficient tumor cells, we developed a robust and sensitive image-based screen using newly generated stable cell lines of HCT116 p53$^{+/+}$ and p53$^{-/-}$ expressing histone 2B (H2B)-GFP and histone 2B (H2B)-RFP fusion proteins, respectively (Fig. 1a)[30]. The cell lines were screened in co-cultures and the relative tolerance of each cell line to the compounds was assessed using automated image acquisition and quantitative analyses of the surviving GFP- and RFP-positive cells.

We identified niclosamide as the most potent compound selective against the growth of p53-deficient cells (Fig. 1b and Supplementary Table 1), validated by viability and apoptotic assays (Fig. 1c–e). Knockdown of p53 using short hairpins against p53 (Supplementary Figure 1a) reduced colony growth (Fig. 1f) and promoted apoptosis in response to niclosamide (Fig. 1g). Next, we extended our observations to isogenic primary human fibroblasts expressing p53 shRNAs or a non-targeting shRNA control. A low dose of 2 μM of niclosamide was sufficient to kill the majority of the cells with knockdown of p53, with minimal effects on the control cells, strengthening the evidence that p53 is a critical determinant of niclosamide response (Fig. 2a and Supplementary Figure 1b). Interestingly, niclosamide activated a marked G1 cell cycle arrest regardless of p53 status (Supplementary Figure 1c), implying that the synthetic lethality is independent of the G1 cell cycle arrest defect in the p53$^{-/-}$ cells.

We further extended our analysis to mouse embryonic fibroblast (MEF). Deletion of p53 in MEFs correlated to increased apoptosis induced by niclosamide (Fig. 2b), as indicated by a high annexin V positivity (>90%) in p53KO MEFs compared to wild-type MEFs and increased cleaved PARP1 and cleaved caspase-3 (Fig. 2c). Moreover, genetic knock-in point mutation p53R172H that mimics the human hotspot p53 mutation (p53R175H) (http://p53.iarc.fr.) also markedly increased apoptosis (Fig. 2d). Notably, mutation of a single allele of p53 (+/R175H) was sufficient to render increased sensitivity to niclosamide compared to wild-type MEFs, while mutation of both alleles (R175H/R175H) greatly exacerbated cytotoxicity even at lower concentrations of the drug (Fig. 2d).

Next, we probed a panel of ovarian and breast cell lines for response to niclosamide. Niclosamide significantly sensitized p53 mutant ovarian (OVCAR-8, TYK-nu) and breast (SKBR3, MDA-MB-231, T47D) cancer cells to niclosamide (Fig. 2e, f), as revealed by a significantly lowered mean IC50 value for the mutant p53 cancer lines compared to the wild-type p53 lines (Fig. 2e). Measurements of cell viability (Fig. 2f), apoptosis (Fig. 2g), and colony growth (Fig. 2h) concurred with the suggestion that loss of p53 either through non-sense or missense mutation predisposes cancer cells to niclosamide. The functionality of the p53 pathway in the ovarian cancer cell lines was confirmed by nutlin-induced induction of p21 or MDM2 in the wild-type p53 cells that was not observed in the mutant cells (Supplementary Figure 1d).

Together, the results provided unequivocal evidence that loss-of-wild-type p53 function, resulting from either inactivating point mutations or a complete loss-of-p53-protein, sensitizes normal and cancer cells of different species and tissue origins to niclosamide.

**Niclosamide uncouples the mitochondria independent of p53**. Similar to mitochondrial uncoupler carbonyl cyanide 4-(trifluoromethoxy)-phenylhydrazone (FCCP) (Supplementary

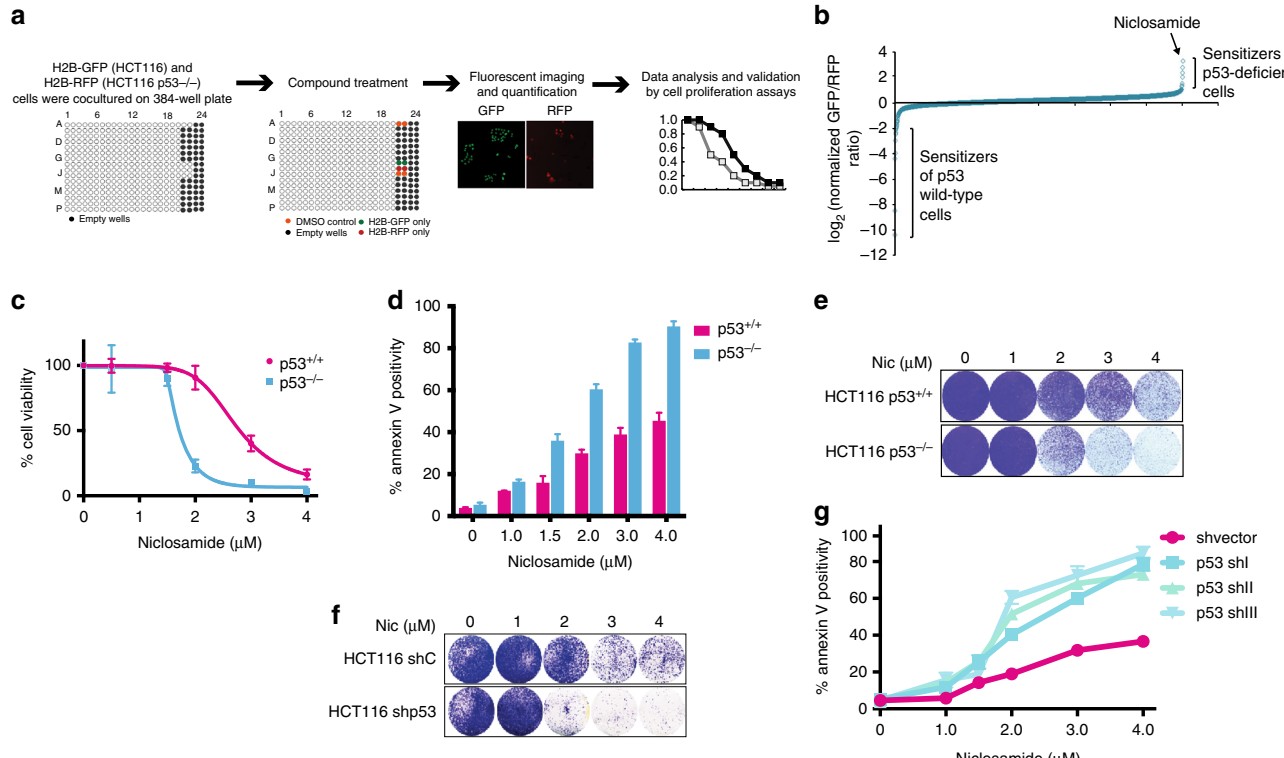

**Fig. 1** p53 deficiency is synthetically lethal with niclosamide. **a** Fluorescent image-based co-culture screen in 384-well using HCT116 p53$^{+/+}$ (H2B-GFP) and HCT116 p53$^{-/-}$ (H2B-RFP) cells. Ratios of percentages of p53$^{+/+}$ (H2B-GFP) and HCT116 p53$^{-/-}$(H2B-RFP) cells normalized to DMSO controls. Log$_2$ values of normalized ratios plotted. **b** Drugs ranked by log$_2$ values of normalized ratios. **c** HCT116 p53$^{+/+}$ and p53$^{-/-}$ cells were treated with niclosamide for 48 h and recovered in drug free media for 4 days. Cell viability measured using WST1 proliferation assay and **d** apoptotic cell death measured using annexin V-FITC conjugated marker in flow cytometry. **e** Crystal violet staining compares survival of HCT116 p53$^{+/+}$ and p53$^{-/-}$. **f** Parental HCT116 were stably transduced with lentiviral constructs expressing shRNA specific to p53 (shp53) or a control shRNA construct. Treatment with niclosamide (48 h) and recovered in drug free media. shp53 cells are significantly more sensitive to niclosamide than NT Sh control cells. Crystal violet staining of shp53 (p53 shI) and NT sh transfected cells and **g** apoptotic cell death were measured using annexin V-FITC marker in flow cytometry. Error bars represent ± SD of at least three independent experiments

Figure 2a, b), niclosamide induced an increase in oxygen consumption even when ATP synthase is inhibited, a hallmark of mitochondrial uncoupling (Fig. 3a–c). Maximal oxygen consumption rates are however similar in wildtype and p53 knockout cells, indicating that both cell lines had undergone similar extent of drug-induced uncoupling (Fig. 3d). Furthermore, using TMRE dye, we observed a reduction in the mitochondrial potential in HCT116 and A549 cells independent of p53 status (Supplementary Figure 2c, d). As expected, a sharp drop in intracellular ATP concentration and an increase in ADP:ATP ratio (Fig. 3e), was observed in both p53$^{+/+}$ and p53$^{-/-}$ cells, albeit to a greater extent in p53$^{-/-}$ cells.

The action of niclosamide in sensitizing p53 knockout cells is due to its activity as a protonophore, since an analogue of niclosamide that contains a methyl (-CH$_3$) group instead of a phenolic hydroxyl (-OH) group (Fig. 3a, f) did not uncouple the mitochondria (Fig. 3g) and had little or no effect on the growth of either wildtype or p53-deficient cells even at high micromolar concentrations (Supplementary Figure 2e, f). Together, our data suggest that niclosamide action in sensitizing p53-deficient cells is intricately linked to its function in mitochondrial uncoupling.

**p53-deficient cells undergo cytochrome c dependent apoptosis.** Niclosamide promoted p53 stabilization and activated canonical p53-dependent transactivation functions (Fig. 3h–j). Absence of p53 increased caspase-9/caspase-3 and PARP1 cleavage in p53$^{-/-}$ cells (Fig. 3k), and was also correlated to mitochondrial

dysfunction and cytochrome c release from the mitochondria in response to niclosamide, as shown by western blot (Fig. 3l) and immunofluorescence (Fig. 3m). The results are consistent with the suggestion that a programmed mitochondrial death pathway comprising of the reported apoptosome cytochrome /APAF1/Cas-9[31–33] may be activated in p53-deficient cells in response to niclosamide, potentially leading to an irreversible apoptotic signaling cascade targeting caspase-3 and PARP1 (Fig. 3k–m).

Niclosamide is reported to inhibit multiple cell regulatory pathways governed by mTOR, STAT3, Wnt, and Notch[21,29]. However, none of these pathways could account for the selective killing of p53-deficient cells by niclosamide, since specific inhibitors to these pathways suppressed growth of p53$^{+/+}$ and p53$^{-/-}$ cells to similar extents, unlike niclosamide (Supplementary Figure 3a–g). Furthermore, inhibition of mTOR and AMPK signaling (Supplementary Figure 3h) and the induction of autophagy, a catabolic process that is inhibited by mTORC1, was also comparable in p53$^{+/+}$ and p53$^{-/-}$ cells (Supplementary Figure 3i). These results prompted us to identify another mechanism in which niclosamide acts to elicit a specific apoptotic response in p53-deficient cells.

**Alteration in metabolome profile imposed by p53 loss.** Although niclosamide disrupts OXPHOS, its effects on the metabolic landscape of cells are not well studied. We performed an untargeted metabolomics profiling of cells treated with niclosamide and a comparative analysis of the metabolomes of

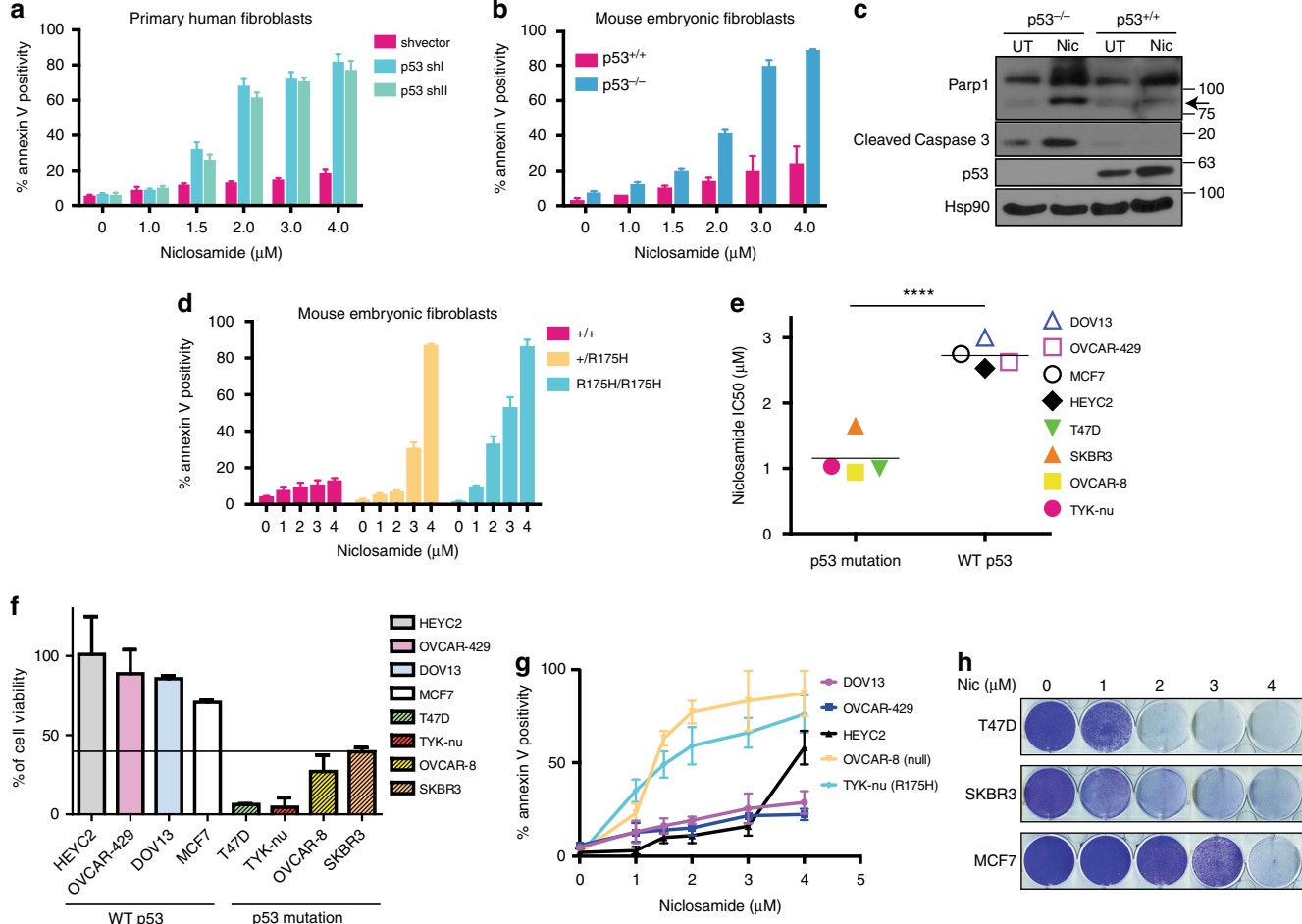

**Fig. 2** Deletion or mutation of p53 results in overt sensitivity to niclosamide. **a** Primary human fibroblasts stably transduced with lentiviral constructs expressing shRNA specific to p53 (p53 shI or p53 shII) or a shvector as control and treated with Niclosamide (48 h). Annexin V positivity measured by flow cytometry. **b** Mouse embryonic fibroblasts (MEFs) from littermates wildtype and p53KO mice treated with niclosamide and annexin V positivity measured by flow cytometry. **c** Cleaved caspase 3 and cleaved PARP1 detected in whole-cell lysate after 24 h niclosamide treatment. **d** Graph represents percentages of annexin V positive cells in niclosamide-treated MEFs from littermate wildtype, heterozygous+/p53R172H and homozygous p53R175H mice. **e** Ovarian and breast cancer cell lines containing p53 mutation or wild-type p53 treated with niclosamide. IC50 values plotted. Statistical paired $t$-test performed; ****$p < 0.0001$. **f** Ovarian and breast cell lines subjected to the indicated concentrations of niclosamide for 48 h and recovered in drug free media for 4 days. Cell viability measured using WST1 proliferation assay and **g** apoptotic cells detected using Annexin V-FITC assay kit by flow cytometry. **h** Colony forming assay of breast cancer cells lines treated with different concentrations of niclosamide. Error bars represent ± SD of at least three independent experiments

drug-treated wildtype and p53 mutant cells. Lysates from DMSO or niclosamide-treated isogenic mouse embryonic fibroblasts (MEFs), wildtype or p53R175H mutant, were subjected to tandem liquid chromatography–mass spectrometry analysis. Over 80 differential analytes pre- and post-niclosamide treatment, including acylglycerols, fatty acids, TCA cycle intermediates, amino acids, and redox intermediates were identified (Supplementary Figure 4a). Principal component analysis (PCA) plots reflected generally similar global metabolic changes triggered by niclosamide, independent of p53 status (Supplementary Figure 4b). For example, we noted a significant decrease in the levels of citric acid, an intermediate in the TCA cycle, as well as energy intermediates such NADP in both wildtype and p53R175H MEFs (Supplementary Figure 4a). However, detailed analysis of the metabolic profiles revealed a significant enrichment of specific fatty acids, in particular, arachidonic acid (AA) (20:4 (ω-6)), eicosatetraenoic acid (EPA) ((20:5 (ω-3)) and docosatetraenoic acid (22:4 (ω-6)) (Fig. 4a, b) and lipid metabolites, lysophosphatidylcholines (LysoPCs) and lysophosphatidylethanoamines (LysoPEs) (Fig. 4c) in drug-treated p53R175H cells compared to

wild-type cells. Consistently, the levels of arachidonic acid was also significantly higher in HCT116 p53$^{-/-}$ than in p53$^{+/+}$ cells post-treatment with niclosamide (Fig. 4d and Supplementary Figure 4c).

**Niclosamide perturbs Ca$^{2+}$ homeostasis**. We reasoned that niclosamide may induce phospholipid hydrolysis to cause an increase in arachidonic acid, LysoPCs and LysoPEs, which are constitutents of phospholipids[34,35]. A majority of phospholipases is activated by Ca$^{2+}$-dependent translocation to the phospholipid membrane[36–38]. Therefore, we first asked if niclosamide causes any changes in intracellular cytosolic Ca$^{2+}$ concentration.

Live cell calcium imaging was monitored using Fluo-4 AM dye (Supplementary Movies 1-6). Rapid calcium fluxes observed in niclosamide-treated HCT116 p53$^{+/+}$ and p53$^{-/-}$ cells were similar in extent; whereas, the niclosamide analog, similar to DMSO controls, did not induce any significant changes in fluorescence (Fig. 5a, b and Supplementary Figure 5a, b). This suggested that niclosamide-induced mitochondrial uncoupling triggers intracellular calcium flux, independently of p53.

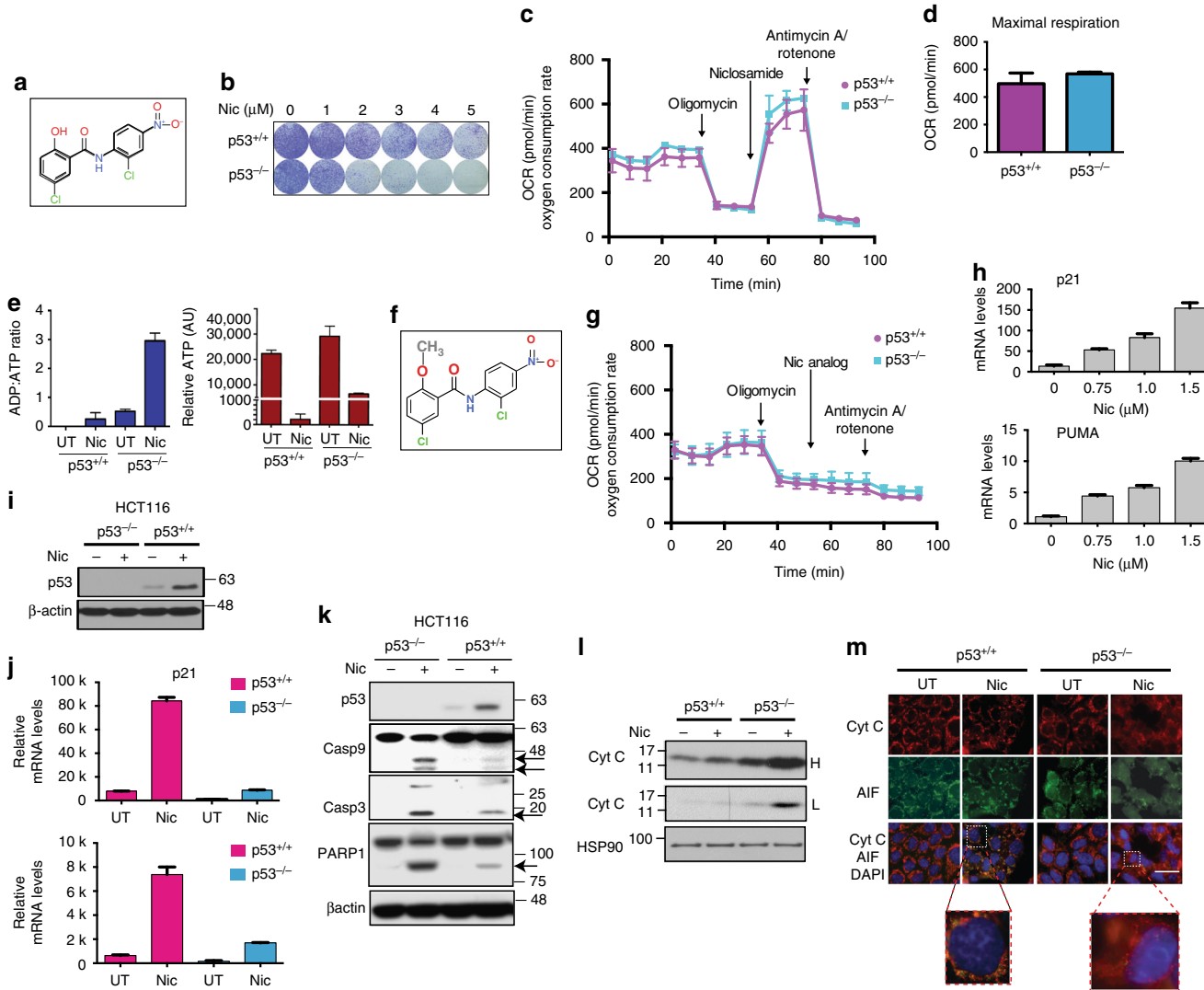

**Fig. 3** Niclosamide induces mitochondrial uncoupling independent of p53. **a** Chemical structure of niclosamide. **b** Crystal violet staining of HCT116 p53[+/+] and p53[−/−] cells treated with niclosamide. **c** Oxygen consumption rate (OCR) measured as a function of time using an extra cellular flux analyzer (Seahorse Bioscience). Seahorse™ Mito Stress Assay was used to measure bioenergetics parameters, by adding ATP synthase inhibitor, oligomycin A, and niclosamide was used in place of FCCP to induce mitochondrial uncoupling. **d** Maximal respiration stimulated by the addition of niclosamide was calculated. **e** Intracellular ADP and ATP measured using ApoSENSOR™ ADP/ATP ratio assay kit and graphs represent the ADP:ATP ratios or the relative ATP levels of HCT116 p53[+/+] and p53[−/−] cells mock-treated or treated with niclosamide. **f** Chemical structure of niclosamide analog. Methyl (-CH3) group replaces the phenolic hydroxyl group in niclosamide. **g** As in **c**, the mitochondrial uncoupling capacity of Nic analog was tested using Seahorse™ Mito Stress Assay. **h** Activation of canonical p53-dependent genes upon niclosmide treatment in a dose-dependent manner (upper panel : p21, lower panel: PUMA). **i** p53 protein detected in whole-cell lysates (WCL) of niclosamide-treated HCT116 p53[+/+] and p53[−/−] cells (**j**) qPCR detection of transcript levels of canonical p53-dependent genes, *p21*, and *PUMA* are shown. **k** Cleavage of caspases 9 and 3, and PARP1 in niclosamide-treated HCT116 cells detected in WCL. **l** Cytosolic fractions of HCT116 p53[+/+] and p53[−/−] cells are immunoblotted for cytochrome c protein. High (H) and low (L) exposures shown. **m** Cytochrome c and apoptosis inducing factor (AIF) detected in fixed cells by immunofluorescence. Scale bar 50 μM. Error bars represent ± SD of at least three independent experiments

To further determine whether mitochondrial uncoupling is an important biochemical function of niclosamide related to its effects in sensitizing p53-deficient cells, we tested another known mitochondrial uncoupler, FCCP. FCCP also sensitized HCT116 p53[−/−] cells to a greater extent (Supplementary Figure 5c, d). Similar to niclosamide, FCCP rapidly induced intracellular calcium fluxes to a similar extent in both HCT116 p53[+/+] and p53[−/−] cells (Supplementary Figure 5e–h; Supplementary Movies 7 and 8). An unbiased metabolome analysis performed in

HCT116 cells revealed significantly a greater fold enrichment in arachidonic acid in HCT116 p53[−/−] cells than in p53[+/+] cells post-treatment with FCCP (Supplementary Figure 5i). Together, our results are consistent with our model that mitochondrial uncoupling affects calcium homeostasis which in turn alters AA metabolism, and potentially explain the hitherto unreported link between niclosamide and the release of arachidonic acid from phospholipids. Furthermore, our data suggest that, whereas, the initial drug-induced calcium flux is independent of p53 status,

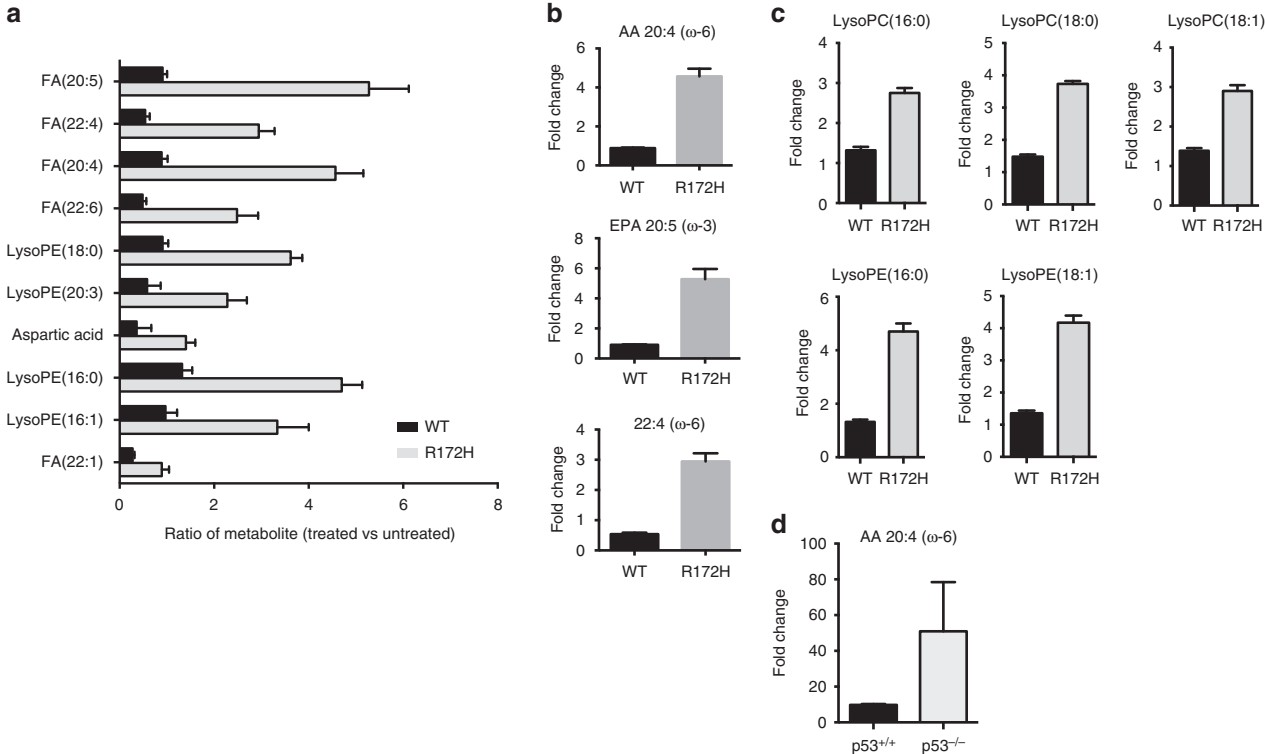

**Fig. 4** Increase in arachidonic acid level in p53-deficient cells. **a** Top 10 metabolites significantly differential between wild-type p53 and mutant p53 MEFs post niclosamide treatment ($p < 0.01$). Paired $t$-test used for statistical analysis. Relative levels of metabolites as measured by liquid chromatography–mass spectrometry (LC–MS) represented as ratio of treated over untreated controls. Data presented as mean ± SD of $n = 9$. **b** Graphs represent the top 3 metabolites differential between drug-treated wildtype and p53 mutant MEFs; fatty acids arachidonic acid (AA;20:4), eicosapentaenoic acid (EPA; 20:5) and docosatetraenoic acid (22:4). Data presented as mean ± SD. **c** Graphs represent two other classes of lipid metabolites (LysoPCs and LysoPEs) differential between drug-treated wildtype and p53 mutant MEFs. A select number of lysoPCs and lysoPEs are presented. Data represent mean ± SD of $n = 9$. **d** Relative level of arachidonic acid is increased in niclosamide-treated HCT116 p53$^{-/-}$ cells compared with treated HCT116 p53$^{+/+}$ cells

however, the steady-state levels of AA is clearly dependent on p53 status, raising the possibility that the turnover of AA is affected in p53-deficient cells.

**Arachidonic acid induces apoptosis in p53-deficient cells**. Next, we hypothesized that the accumulation of arachidonic acid (AA) in p53-deficient cells may ultimately lead to a programmed cell death. To directly test this, we exposed cells to exogenous AA. AA induced a dose-dependent increase in apoptosis (Fig. 5c, d) and reduced colony growth more in p53-deficient cells compared to wild-type cells (Fig. 5e). Moreover, AA-induced cytochrome c release to a greater extent in p53-deficient cells than wild-type cells, mimicking the effects of niclosamide treatment (Fig. 5f). Together, our data suggested that AA and niclosamide induce apoptosis potentially through a common pathway involving cytochrome c release from the mitochondria, consistent with the suggestion that niclosamide mediates cell death partly through the effects of arachidonic acid. Interestingly, AA has been found to increase mitochondrial permeability transition (PT) and cytochrome c release[39]. Importantly, similar to niclosamide, we demonstrated that this phenomenon is further exacerbated upon p53 loss.

We demonstrated that inhibition of cPLA2 (cytosolic phospholipase A2), using two-specific inhibitors of cPLA2, suppressed cytosolic cytochrome c induction by niclosamide in p53$^{-/-}$ cells (Fig. 5g) and rescued partially the growth of cells challenged with niclosamide (Fig. 5h). The PLA2 inhibitors on their own did not significantly or adversely affect cell viability at the concentrations tested (Supplementary Figure 5j). Together, our results concur

with the suggestion that niclosamide induces cytochrome c release and apoptosis, at least in part, through AA release.

In line with the above suggestion that niclosamide may function through calcium-dependent cPLA2 activation and AA release, partly through calcium signaling, we tested how a commonly used intracellular chelator of calcium ions, BAPTA, may affect niclosamide effects. Cells reloaded with BAPTA-AM showed a significantly reduced niclosamide-induced calcium response (Supplementary Figure 5k(i)). When combined with another calcium chelator, EGTA, niclosamide-induced calcium levels were further reduced (Supplementary Figure 5k(ii)). BAPTA-AM also significantly reduced the extent of PARP1 cleavage induced by various concentrations of niclosamide in p53-deficient cells (Supplementary Figure 5l) and, in combination with EGTA, significantly reduced the extent of annexin V positivity (Supplementary Figure 5m). Together, our data concur with the suggestion that calcium flux plays a role in mediating the observed cytotoxic effects of niclosamide, at least in part.

In agreement with the above results, we further demonstrated that carbacyclin, a prostacyclin analogue that is shown to attenuate calcium levels by inhibiting calcium release from intracellular stores[40,41], antagonized niclosamide-induced growth inhibition and restored partially the growth of cells challenged with niclosamide (Supplementary Figure 5n). Carbacyclin alone did not have significant adverse effects on cell viability (Supplementary Figure 5o, p).

**New p53 target gene identified in lipid oxygenation pathway**. Our data led to the suggestion that p53 may play a major role in the metabolism of arachidonic acid in cells. Therefore, we

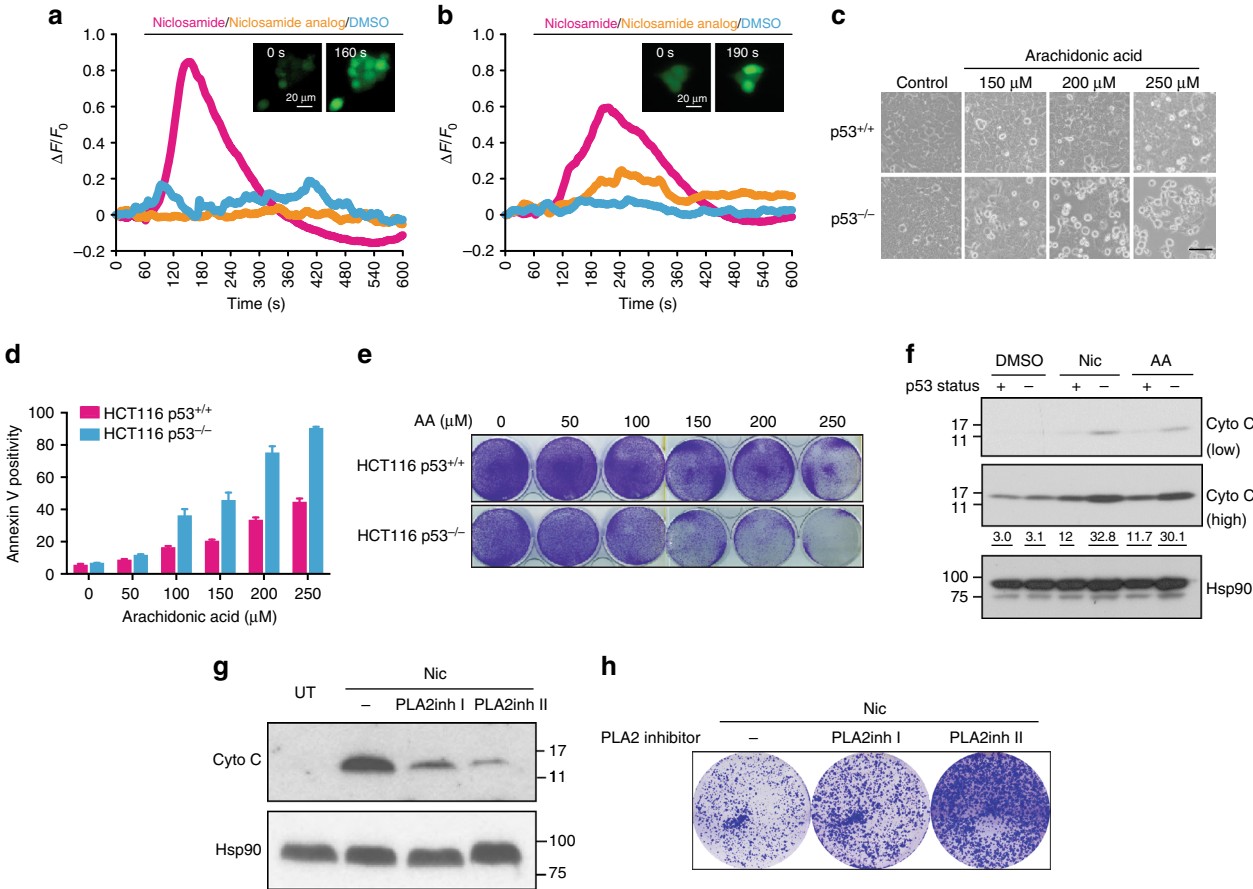

**Fig. 5** Niclosamide alters intracellular calcium flux. **a**, **b** Calcium flux was monitored by live cell imaging using Fluo-4, AM dye. Cells were loaded with Fluo-4, AM calcium indicator for 30 min before addition of niclosamide. Changes in Fluo-4 fluorescence intensity of cells were measured and quantified using Image J. Normalized values divided by initial fluorescence ($\Delta F/F_O$) were plotted. Inactive niclosamide analog and DMSO were included as controls. Each line represents the average of Fluo-4 fluorescence intensities obtained from single cells. Representative fluorescence images of HCT116 cells before and after treatment with niclosamide show an increase in intracellular calcium. Experiments are repeated at least three independent times. **c** Treatment with arachidonic acid (in Tocrisolve$^{TM}$100) results in increased cell death in HCT116 p53$^{-/-}$ cells compared to HCT116 p53$^{+/+}$ cells. Bright field images are shown. Scale bar 50 µM. Control refers to treatment using a water soluble emulsion Tocrisolve$^{TM}$100. **d** Apoptotic cells measured using annexin V-FITC marker by flow cytometry. Data represent mean ± SD of at least three independent experiments. **e** Crystal violet staining of HCT116 p53$^{+/+}$ and p53$^{-/-}$ cells treated with arachidonic acid. **f** HCT116 p53$^{+/+}$ and p53$^{-/-}$ cells treated as in **c** are fractionated and the cytosolic fractions are immunoblotted using an antibody specific to cytochrome c. High and low exposures shown. Numbers below the immunoblot represent densitometric measurements of the band intensities. **g** Inhibitors of PLA2 (PLA2inh I and PLA2inh II) were added in combination with niclosamide (2 µM) to HCT116 p53$^{-/-}$ cells and cytosolic cytochrome c was detected by immunoblotting. Treatment with niclosamide (2 µM) alone was included. **h** Drug-treatment conditions were similar to **g**; surviving colonies of cells were stained using crystal violet following recovery from 48 h drug treatment

examined if p53 regulates any of the 18 genes from the three major pathways, namely, lipoxygenases (LOXs), cyclooxygenases (COXs), and cytochrome P450 epoxygenase (EPOs) that converts AA to leukotrienes, prostaglandins, and thromboxanes (Fig. 6a and Supplementary Table 2). Remarkably, we found that the transcripts of two unique genes, arachidonate 5-lipoxygenase (*ALOX5*)[42] and arachidonate 12-lipoxygenase,12R type (*ALOX12B*), were most significantly enriched in doxorubicin-treated HCT116 and A549 cells in a p53-dependent manner, similar to *p21*, a canonical p53 target gene (Fig. 6a–c and Supplementary Figure 6a). By contrast, the changes in mRNA expressions of other *LOXs*, *COXs*, or *EPOs* genes after doxorubicin were comparable in the isogenic HCT116 cells (Fig. 6a and Supplementary Figure 6b).

Niclosamide induced the transactivation of canonical p53 targets and stabilization of p53 protein (Figs. 3h–j, and 6d). Similar to doxorubicin, niclosamide also led to a robust induction of *ALOX12B* and *ALOX5* genes, almost comparable to the extent of *p21* induction, in a p53-dependent manner (Fig. 6e). To further

test whether these genes are also induced in response to a more specific inducer of p53 activity, we used an MDM2 antagonist, Nutlin[43], which blocks the binding of MDM2 to p53, resulting in p53 stabilization and activation. Gene and protein expressions of ALOX5 and ALOX12B were specifically induced by Nutlin in wild-type HCT116 cells but not in p53$^{-/-}$ cells, as were expected of p53-induced genes (Fig. 6f and Supplementary Figure 6c). Next, to assess how generally these genes are activated by p53 and niclosamide, we tested the transcriptional induction of these genes in MEFs. Consistently, normal wild-type MEFs displayed a clear induction of *Alox5* and *Alox12b* gene expression following niclosamide treatment, in contrast to p53R172H MEFs (Fig. 6g). Evidently, these are bona-fide p53 targets even in vivo; similar to *p21* and *Mdm2*, a significant reduction in transcript levels of *Alox5* and *Alox12b* was detected in kidney tissue derived from the p53KO and p53R175H mice when compared to the wild-type littermates (Supplementary Figure 6d–e).

Finally, to determine whether these genes are direct p53 targets, we first examined the interaction of p53 with the genomic loci of

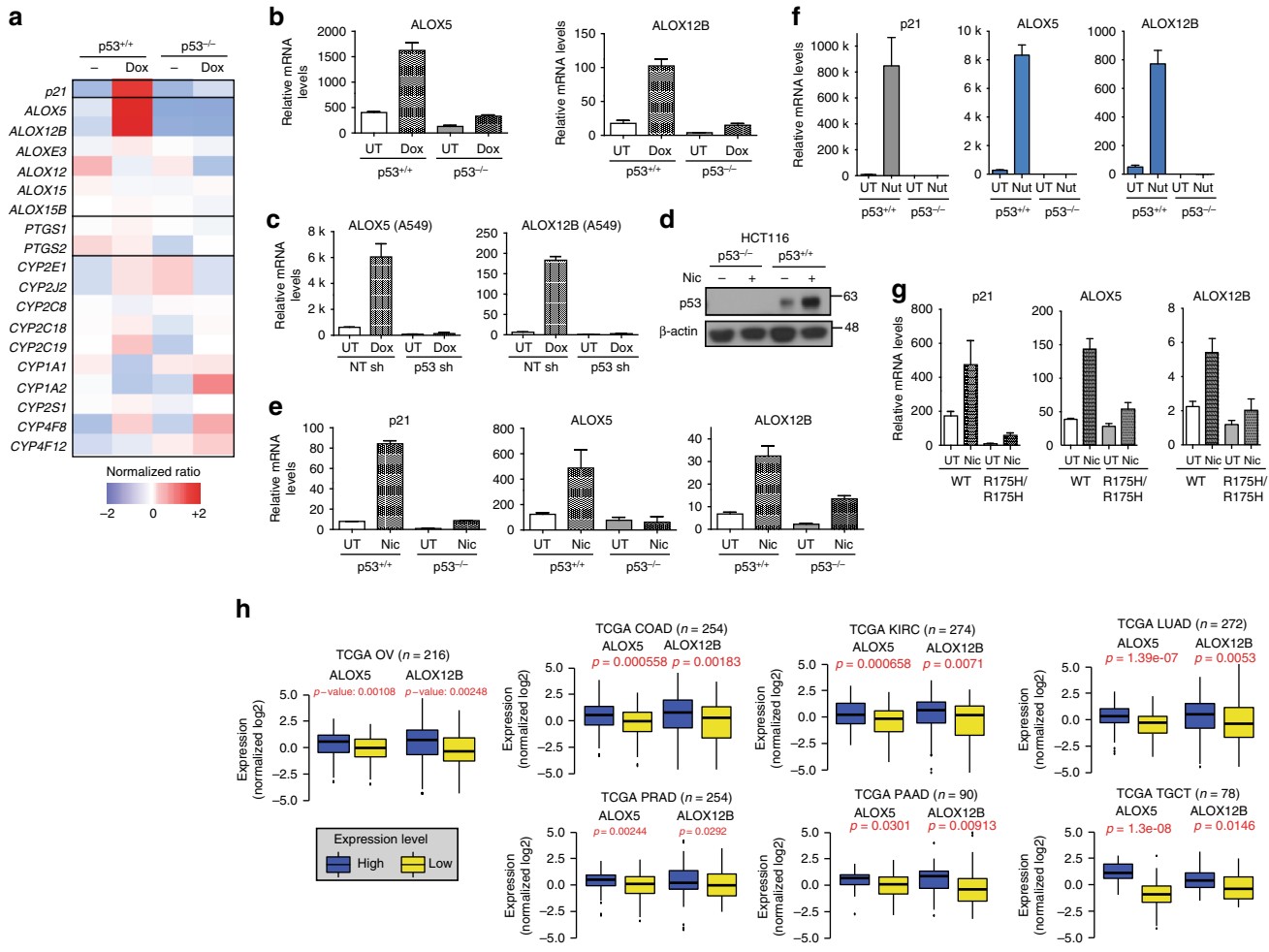

**Fig. 6** p53 promotes transcription induction of ALOX5 and ALOX12B. **a** LOX (lipoxygenase), COX (Cyclooxygenase), and EPO (epoxygenase) genes are screened for their dependency on p53 for expression. Heatmap summarizes the relative fold changes of transcript levels of genes in doxorubicin (Dox)-treated HCT116 p53$^{+/+}$ and p53$^{-/-}$ cells. **b** Relative transcript levels of *ALOX5* and *ALOX12B* in doxorubicin (Dox)-treated (0.2 μM) HCT116 p53$^{+/+}$ and p53$^{-/-}$ cells measured by qPCR. **c** Lung adenocarcinoma A549 cells stably transfected with p53-specific shRNA (p53 shI) or control shRNA treated with 0.2 μM doxorubicin (16 h). **d** p53 protein stabilized by niclosamide. **e** Relative transcript levels of *p21, ALOX5*, and *ALOX12B* genes in HCT116 p53$^{+/+}$ and p53$^{-/-}$ cells treated with niclosamide (24 h) or **f** with nutlin (24 h). **g** MEFs from littermates wildtype and p53R175H mice treated with niclosamide (24 h) and relative transcript levels of *p21, ALOX5*, and *ALOX12B* genes measured by qPCR. **h** Gene expression profiling of *ALOX5* and *ALOX12B* genes in TCGA RNA-seq datasets. Samples ranked accordingly based on the *p53* gene signature (Supplementary Methods, Supplementary Figure 6g–m)[48]. Normalized log$_2$ values for *ALOX5* and *ALOX12B* were plotted in a graph and *p*-values indicated (unpaired *t*-test). TCGA OV (ovarian) (*n* = 216), TCGA COAD (colon adenocarcinoma) (*n* = 254), KIRC (kidney renal clear cell carcinoma) (*n* = 274), LUAD (lung adenocarcinoma) (*n* = 272), PRAD (prostate adenocarcinoma) (*n* = 254), pancreatic adenocarcinoma (PAAD) (*n* = 90) and testicular germ cell cancer (TGCT) (*n* = 78). Error bars represent ± SD of at least three independent experiments

ALOX5 and ALOX12B using bioinformatics analysis of established p53 ChIP-seq database (GSM1142696[44]). We visualized distinct p53 binding peaks on *ALOX12B* and *ALOX5* in the UCSC Genome Browser using reference genome hg38 (Supplementary Figure 6f). To further verify whether these are p53 binding sites, we performed ChIP experiments. We incubated sonicated lysates from HCT116 p53$^{+/+}$ and p53$^{-/-}$ cells with a p53-specific antibody (DO-1) and examined the binding of p53 to *ALOX5* gene locus using site-specific qPCR primers that amplify the putative p53 peak region or sites (A and B) that do not appear to binding p53. qPCR analyses revealed that p53 specifically binds to the region corresponding to the putative p53 peak but not to the irrelevant regions A and B (Supplementary Figure 6g). Similarly, ChIP-qPCR analyses revealed a specific enrichment of p53 binding to Peak I and II on *ALOX12B* gene locus, but not to the irrelevant region C (Supplementary Figure 6g). Together, the data demonstrate that p53 binds specifically to *ALOX5* and *ALOX12B*

gene loci within intragenic regions and suggested that *ALOX5* and *ALOX12B* are direct gene targets of p53. The binding of p53 to intragenic regions is consistent with reports suggesting that p53 can bind to enhancer sites away from promoter regions to promote transcription activation[45,46]. Taken together, the results indicated that *ALOX5* and *ALOX12B* genes are p53 targets induced in response to acute genotoxic and metabolic stress.

**ALOX5/12B expression correlates to wild-type p53 in TCGA data.** To ask whether the gene expression of *ALOX5* and *ALOX12B* is related to the functional status of p53 in human tumors, we conducted a gene-profiling analysis using large scale RNA-seq transcriptomic datasets from The Cancer Genome Atlas (TCGA) (https://cancergenome.nih.gov/). We stratified the global gene expression profiles according to a *p53* gene signature[47,48] that could be a more accurate measure of p53 functionality. We established a scoring system (Materials and Methods) to rank the

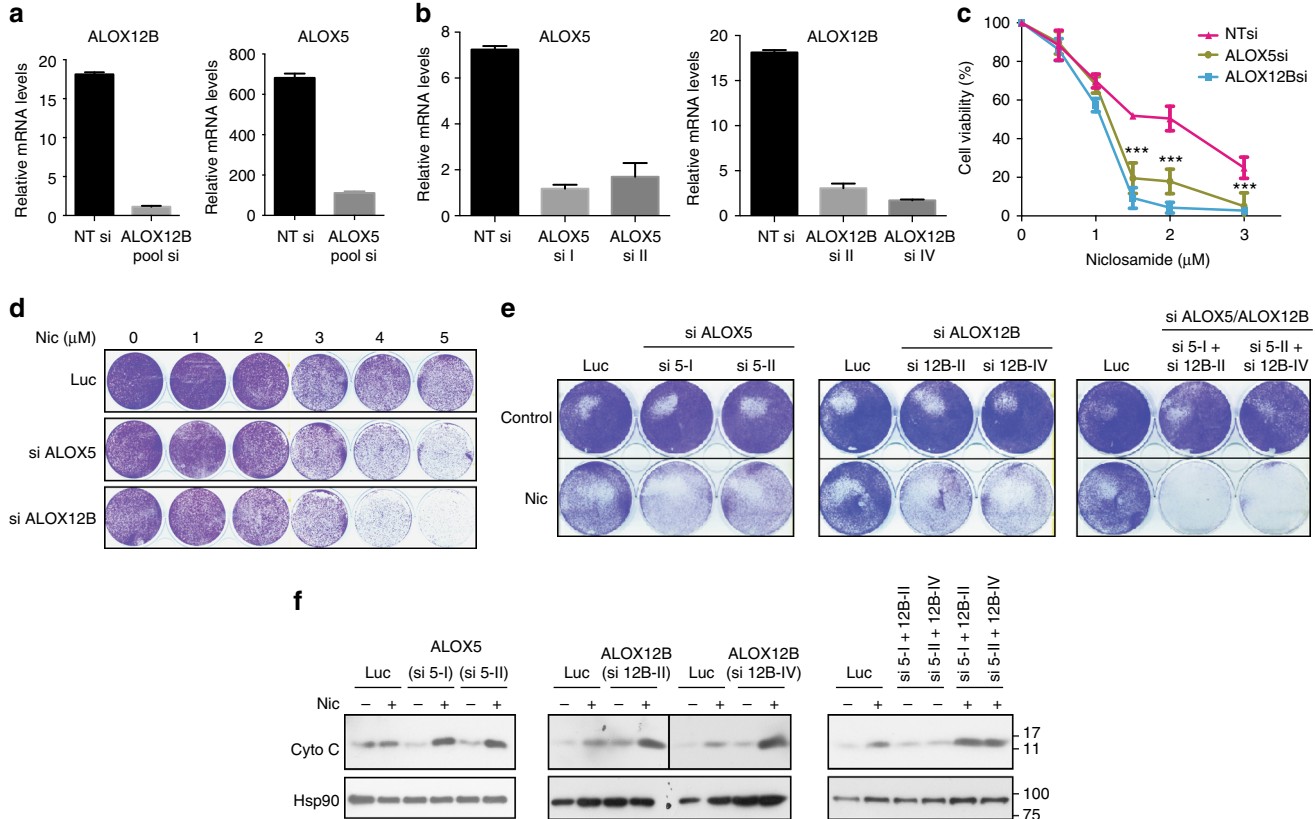

**Fig. 7** p53-induced activation of ALOX5 and ALOX12B counteracts niclosamide. **a** Knockdown of *ALOX5* and *ALOX12B* gene expression by Dharmacon SMARTpool® siRNAs assessed by qPCR. **b** Knockdown of *ALOX5* and *ALOX12B* gene expression by individual siRNAs measured by qPCR. **c** Reduction in cell viability of HCT116 cells transfected with either SMARTpool® siRNAs targeting *ALOX5* and *ALOX12B* compared to cells transfected with NT (non-targeting) siRNAs. **d** Knockdown of *ALOX5* and *ALOX12B* sensitizes HCT116 cells to niclosamide (1.5 µM). Crystal violet staining of cells shown. **e, f** HCT116 cells transfected with individual siRNAs targeting *ALOX5* (si I and si II), *ALOX12B* (si II and si IV), or a combination of *ALOX12B* siRNA and *ALOX5* siRNA, and treated with niclosamide (24 h) were examined for cell survival in colony assays at the end of 8 days post recovery from drug treatment and cytosolic cytochrome c levels. Luc siRNA included as control. Hsp90 as loading control. Error bars represent ± SD of at least three independent experiments

samples. Upper and lower quartiles were classified as "high" and "low" *p53* gene signature cohorts. The unbiased approach showed that expression of *ALOX5* and *ALOX12B* are significantly higher in "high" expressing group, compared to the "low" expressing group, in the TCGA ovarian tumor cohort ($n = 216$; *ALOX5 p = 0.00108* and *ALOX12B p = 0.00248*) as well as in other tumor cohorts (Fig. 6h and Supplementary Figure 6h–n).

**p53-induced activation of *ALOX5 /12B* counteracts niclosamide**. To gain additional insights into the pro-survival function of p53 activated by niclosamide, we tested whether the p53-dependent genes, *ALOX5* and *ALOX12B*, are essential for cell survival after niclosamide treatment. Efficient knockdown of *ALOX5* or *ALOX12B*, either using pooled (Fig. 7a) or individual siRNAs (Fig. 7b), clearly reduced cell viability and colony growth in response to niclosamide indicating that both genes are critical for the prosurvival function of p53 in this context (Fig. 7c–e). Decrease in colony growth was also correlated to a significant increase in cytosolic cytochrome c (Fig. 7f). As expected, knockdown of *ALOX5* or *ALOX12B* barely further compromised the viability of p53$^{-/-}$ cells in response to niclosamide (Supplementary Figure 7). Collectively, these data suggest that p53 exerts its prosurvival function, at least in part, by controlling the transcription of lipid oxygenation genes *ALOX5* and *ALOX12B* to counteract niclosamide-induced metabolic stress.

**Niclosamide inhibits growth of p53 mutant xenografts**. To illustrate the impact of the genetic status of p53 on the efficacy of niclosamide in vivo, we injected isogenic HCT116 p53$^{+/+}$ and p53$^{-/-}$ cells as tumor xenografts in nude mice and performed oral dosing of niclosamide for 28 days. We observed a significant tumor growth delay in HCT116 p53$^{-/-}$ tumors; tumor growth inhibition measured at the end of study is 45.7% (Fig. 8a). In contrast, drug treatment of HCT116 p53$^{+/+}$ tumors only led to a 15.7% tumor growth inhibition compared to control (Fig. 8b). Kaplan–Meier analysis of the tumor growth delay data, and comparison of the niclosamide-treated and vehicle treated groups in the p53KO tumor cohort using log-rank *t*-test analysis showed that both curves are statistically significantly different ($P < 0.05$), in contrast to the p53 wild-type tumor cohort ($P > 0.05$) (Supplementary Figure 8a, b). Consistently, increased cleaved caspase 3 was detected in HCT116 p53$^{-/-}$ tumors sections (Supplementary Figure 8c). To test whether the AA levels in in vivo HCT116 tumors reflects that observed in HCT116 cells in vitro, we harvested the xenografts at the end of the study for LC–MS metabolome detection of AA. However, as shown in Fig. 8c, we detected almost equivalent levels of AA in HCT116 p53$^{+/+}$ and p53$^{-/-}$ tumors, in contrast to what we observed in cells (Fig. 4d). One plausible explanation is that since AA accumulation is linked to niclosamide-induced cell death, we reasoned that those cells with high AA accumulation would have been eliminated at the end of the xenograft study. The remaining tumor cells would probably have the poorest response to niclosamide and therefore

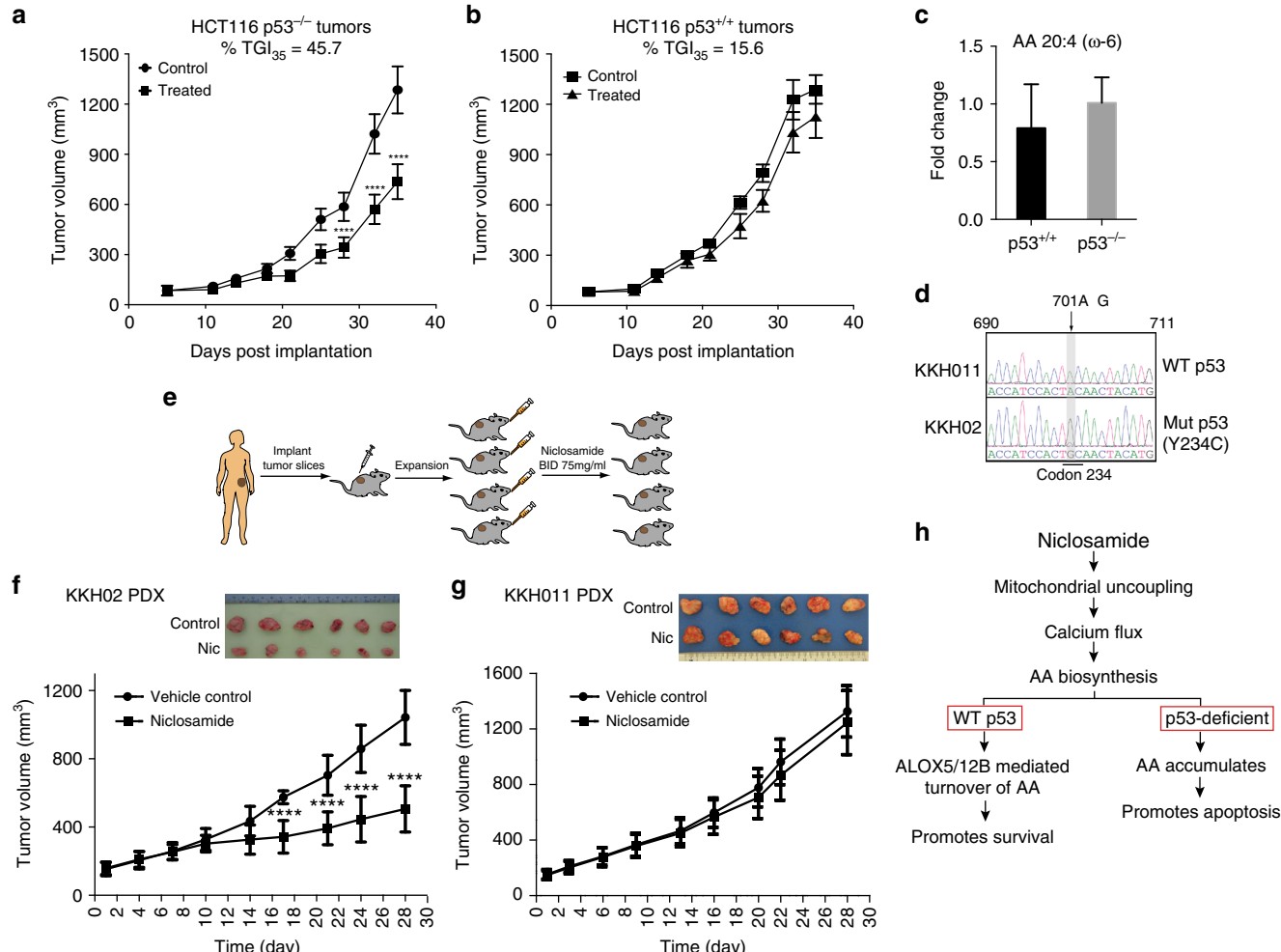

**Fig. 8** p53 loss improves therapeutic outcome of niclosamide treatment. **a**, **b** Tumor growth of HCT116 p53$^{+/+}$ and p53$^{-/-}$ xenografts monitored for the entire duration of experiment. Randomized cohorts of mice ($n = 16$) were subjected to niclosamide or vehicle control twice daily. Data represent mean ± SEM. Unpaired $t$-test was used to determine statistical significance; ****$P < 0.0001$. Percentage tumor growth inhibition (TGI) at the end of study (day 35) as indicated. **c** AA levels measured in HCT116 p53$^{+/+}$ and p53$^{-/-}$ xenografts harvested at the end of the tumor growth study. **d** Direct DNA sequencing of the complete p53 genomic locus was performed for KKH011 and KKH02 PDX. Histogram shows the DNA sequencing trace at position 690–711 bp of *p53* gene. Highlighted in gray box is the point mutation detected in KKH02 (701A>G) that translates to an Y234C mutation in p53. **e** Schematic diagram of the in vivo PDX in SCID mice. Cohorts of PDX models of ovarian cancer (KKH02 and KKH011) were established. When tumors reached an average volume of 100–150 mm$^3$, randomized cohorts of mice ($n = 16$) were subjected to oral gavage with niclosamide (75 mg/kg) or vehicle control twice daily. **f**, **g** Tumor growth of KKH02 and KKH011 PDXes monitored by caliper measurement for the entire duration of experiment. Data represent mean ± SEM. Unpaired $t$-test was used to determine statistical significance; ****$P < 0.0001$. Images show the tumor xenografts dissected at the end of the experiments. **h** Proposed model depicts the molecular mechanism underlying niclosamide action that subsequently leads to sensitization of p53-deficient cells, whereas wild-type p53 cells circumvent the effects of niclosamide by transcriptionally activating *ALOX5* and *ALOX12B* genes

did not accumulate sufficient AA to induce cell death. Additionally, tumors are much more heterogenous in nature compared to cell cultures, with a strong likelihood of host cells (wildtype for p53) vascularizing the xenografts. The presence of host-derived cells that are wildtype for p53 would cause additional complexities in accurately comparing the AA levels.

To further confirm the therapeutic relevance of niclosamide for p53-deficient tumors, we shifted to an in vivo tumor model using patient-derived ovarian tumor explants grown in SCID mice (Supplementary Table 3). We selected a pair of ovarian PDXes that exhibit similar tumor growth kinetics and are paired for p53 wildtype and mutant genotypes. DNA sequencing revealed that KKH011 is wildtype for p53 sequence and KKH02 contains a single point mutation (701A>G) that translates to a Y234C codon mutation found in human cancers (http://p53.iarc.fr) (Fig. 8d).

An intense p53 staining was detected in the mutant p53 tumor (KKH02), in contrast to the wild-type p53 tumor (KKH011), consistent with a characteristic stabilization of mutant p53 protein reported in tumors[49] (Supplementary Figure 8d). Mice bearing xenografts were treated with vehicle, or niclosamide at 75 mg/kg twice daily for 4 weeks (Fig. 8e).

We observed a significant reduction in the growth of KKH02 tumors (p53Y234C), compared to vehicle treated control tumors, with ~50% tumor growth delay (Fig. 8f), in sharp contrast to the KKH011 PDX model (Fig. 8g). Histological examination of both PDX tumors revealed an obvious reduction in the number of Ki67 positive cells, indicative of a drug-induced proliferative arrest, and suggested that both tumors were exposed to niclosamide at the oral dosage administered (Supplementary Figure 8d). In treated KKH02 tumors, this was accompanied by

an increase in cleaved caspase 3 positivity (Supplementary Figure 8d). Taken together, our results indicate that loss-of-wild-type p53 functions selectively confers improved therapeutic response to niclosamide.

## Discussion

Here we described the use of niclosamide as a single agent in simultaneously activating p53 pro-survival function while rendering acute metabolic stress and cytotoxicity in p53 mutant/knockout cells. We found that p53 controls the cell fate decision by reversibly activating lipid oxygenation genes *ALOX5* and *ALOX12B* in response to an acute dose of niclosamide (Fig. 8h). The substrate of ALOX5 and ALOX12B, arachidonic acid, is central to niclosamide-induced metabolic stress. Loss-of-wild-type p53 activity results in a marked accumulation of arachidonic acid and increase in cytochrome c release, caspase 9/3 activation and apoptosis (Fig. 8h). Importantly, the selectivity of niclosamide for p53-deficient cells is recapitulated in in vivo tumor models, resulting in increased anti-tumorigenic effects on p53-mutant tumors compared to wild-type p53 tumors in ovarian patient-derived xenograft models.

We described a new p53-ALOX5/12B axis that protects wild-type cells from niclosamide. Our data point to a new mechanism of niclosamide action dependent on its mitochondria uncoupling activity in inducing an intracellular calcium flux. Increased intracellular $Ca^{2+}$ triggers the activation of many enzymes including those that hydrolyzes phospholipids, such as phospholipase A2[50], leading to the release of arachidonic acid (Fig. 8h). Evidently, we showed that calcium chelators or inhibitors of intracellular calcium release as well as cytosolic PLA2 inhibitors attenuate the effects of niclosamide. Consistent with this, metabolome profiling revealed an increase in components of phospholipids, namely, LysoPEs and LysoPCs, which are released in addition to arachidonic acid upon hydrolysis of the sn-2 acyl bond of phospholipid by phospholipase A2. This pathway provides precursors for the generation of leukotrienes, thromboxanes, and prostaglandins, etc, through the action of a multitude of enzymes that subsequently metabolize arachidonic acid[35]. Our data suggest that the presence of other AA metabolizing enzymes in p53-deficient cells is clearly not sufficient to prevent the catastrophic accumulation of intracellular AA. The imbalance in the rates of generation and turnover of AA is prevented however in wild-type cells by a significant drug-induced p53-dependent transcriptional activation of *ALOX5* and *ALOX12B* genes. Through knockdown experiments, we have dissected the key components linking p53 activation to niclosamide sensitivity and identified *ALOX5* and *ALOX12B* as important modulators of niclosamide response. Together, our findings place p53-ALOX5/12B axis central to the regulation of metabolic homeostasis and pro-survival functions after niclosamide treatment.

Although, recent reports implied that niclosamide inhibits tumor growth through suppressing Wnt, Notch, mTOR and STAT3 pathways, its exact antitumor mechanism remains unclear. Herein, we identified p53 as a critical determinant of niclosamide response for the first time and demonstrate that even in a Wnt/beta-catenin mutant background in HCT116, knockout of *p53* gene significantly increases apoptosis. The increased niclosamide sensitivity of p53 knockout in a Wnt/beta-catenin mutant background was not because p53 loss somehow alters the Wnt signaling pathway and enhances the effects of Wnt targeting drugs since Wnt inhibitors did not cause any preferential death of p53-deficient cells (Supplementary Figure 3f). Rather, this implies that p53 regulates another mechanism independent of the Wnt pathway, at least in this context, to modulate cellular response to niclosamide. Here we revealed unexpectedly that niclosamide-induced mitochondrial uncoupling affects normal phospholipid biogenesis and activates a homeostatic response mechanism through the p53 pathway.

The most exciting observation pertaining to potential treatment for p53 mutant tumors is that niclosamide regresses p53 mutant tumors more effectively than wild-type tumors, as herein demonstrated using human ovarian PDX mouse models and isogenic cell line xenograft models. Deletion of the *p53* gene in HCT116 p53[−/−] rendered the tumor xenografts significantly more responsive to niclosamide treatment (Fig. 8a, b) and so did mutation of p53 in an ovarian PDX model that we established. This is consistent with the striking increase in potency of niclosamide on MEFs bearing a different p53 point mutation (p53R175H) or a genetic knockout at the *p53* gene locus (p53[−/−]) compared to wild-type cells (Fig. 2b, d). Moreover, because this therapeutic approach targets the loss-of-functional wild-type p53 rather than a specific p53 point mutation or defect, this approach is likely to be effective on a broad spectrum of tumors harboring defects in the p53 pathway, including inactivating point mutations and deletions of *p53* gene, and well tolerated in vivo. Indeed, our animal studies also revealed that niclosamide is well-tolerated and no adverse effects on mice weight and physiology was reported during the course of oral dosing. This is congruent with cell-based studies showing a good tolerability of normal cells to niclosamide and loss of p53 in an untransformed background imposing a substantial effect on niclosamide sensitivity. Together, the results suggest that niclosamide induces a prosurvival mechanism, mediated through p53, which can be exploited to render tumor-specific targeting through the simultaneous activation of p53 and the induction of an arachidonic acid/cytochrome c apoptotic response in tumors driven by the loss of p53.

Importantly, expression of p53-ALOX5/ALOX12B pathway components could be useful for tumor stratification and direct niclosamide treatment to those likely to respond. We have capitalized on the advancements of large scale transcriptomics that allow for the interrogation of extensive tumor data sets to identify and validate new gene–gene relationships, as well as provide a means to differentiate cohorts of cancers accordingly. The TCGA RNA-seq transcriptomics data revealed significant correlation between *ALOX5* and *ALOX12B* gene expression and the wild-type *p53* gene signature[48] (Fig. 6h and Supplementary Figure 6h–n). This provides a compelling suggestion that in addition to the *p53* gene signature, the inclusion of *ALOX5* and *ALOX12B* will provide a clinically annotated gene signature that can potentially be used as a predictor of niclosamide response.

Additionally, other parallel mechanisms, such as the loss of heterozygosity of the *p53* gene with a concomitant deletion of 17p13.1 that is known to drive aggressive tumor phenotypes, could potentially promote arachidonic acid accumulation as a result of an incidental deletion of *ALOX12/15* genes[51] and provide another therapeutic opportunity for niclosamide in the treatment of this subset of aggressive human tumors.

This study provides direct evidence that the deletion/mutation of p53 and the impaired expression of lipid oxygenation genes confer improved therapeutic response to niclosamide. We provide new insights for a previously unknown mechanism of niclosamide action and highlight new exciting therapeutic opportunities for the use of niclosamide in anticancer therapy against p53 mutations.

## Methods

**Screening of drug compounds using p53[+/+] and p53[−/−] cells.** The parental isogenic cell lines used in this study was a kind gift from Prof. Bert Vogelstein Laboratory. This cell line and various other cell lines used later in this study were regularly subjected to mycoplasma test. We have used MycoAlert™ Mycoplasma Detection Kit (Catalog #: LT07-118). HCT116 p53[+/+](H2B-GFP) and HCT116

p53$^{-/-}$ (H2B-RFP) cells were grown in co-culture and screened against a Pharmakon Library in 384-well. Pharmakon compound library was purchased from MicroSource Discovery Systems, Inc. (USA). Briefly, equal numbers of HCT116 p53$^{+/+}$ and HCT116 p53$^{-/-}$ cells were added to 384-well plates containing drugs at 2 μM concentration at final cell density of 400 cells per well in a final volume of 80 μl per well. After 5 days of drug incubation, plates were imaged using an In Cell Analyzer 2000 (GE) in FITC and Texas Red channels. Automated image acquisition followed by quantitative image analysis using InCell Investigator software was performed to quantify the number of GPF and RFP positive cells. Ratios of GFP/RFP were divided by that of DMSO controls and expressed as log$_2$ values.

**Metabolome profiling.** For experiments involving cell lines, $2 \times 10^7$ cells per sample were collected and gently washed with ice-cold 150 mM sodium chloride (Merck, Darmstadt Germany) to quench cellular metabolism. Quenched samples were collected by centrifugation at 3000 g for 5 min at 4 °C. For the experiment with mouse xenografts, the tumor was collected, rinsed with ice-cold 150 mM sodium chloride and lyophilized using a freeze dryer (Alpha 2–4 LD plus, Martin Christ Gefriertrocknungsanlagen, Germany). A quantity of 10 mg per lyophilized sample was subsequently weighed out for metabolite extraction. Metabolites were extracted using a two-phase liquid–liquid extraction protocol, previously described[52]. Briefly, methanol, chloroform, and 3.8 mM tricine (Merck) solution (approx. 1:1:0.5 v/v) was used to separate polar metabolites (aqueous) from lipid species (organic fraction). Polar metabolites in the aqueous fraction were collected in microcentrifuge tubes. Lipid metabolites in the organic fraction were stored in 2 mL amber glass vials and the headspace filled with nitrogen gas to minimize sample degradation. All extracts were stored at −80 °C and reconstituted in the respective chromatography solvents prior to analysis.

**Data processing and analysis for metabolites.** Raw LC–MS data obtained were pre-processed using the XCMS peak finding algorithm[53]. Detected mass peaks were assigned putative metabolite identities by matching the respective masses (<10 ppm error) with the KEGG and Human Metabolome Database (HMDB). Total area normalization was applied to the pre-processed data prior to statistical analysis using multivariate (SIMCA-P+ software) and univariate tools, including relative ratios, Student's t-test (Welch′s correction) and hierarchical clustering for the classification of common trends. Where possible, the metabolite identities were confirmed by MS–MS spectral comparison with commercially available metabolite standards[54].

**Fluorescence live cell calcium imaging (Fluo-4 assay).** A total of 25,000 cells were seeded into 35 mm glass bottom dishes. Next day, cells were incubated with 2 μM Fluo-4, AM dye (Invitrogen, F10489) in serum free medium for 30 min. Live cell imaging was performed using IX83 microscope (Olympus) with FITC filters at 37 °C in 5% CO$_2$. Niclosamide, inactive niclosamide analog or DMSO (0.1%) was added to the media 1 min after the start of live cell imaging. Events were recorded every 10 s. Quantitative analyses were performed using ImageJ. Ca$^{2+}$ changes expressed as $\Delta F/F_0$ values vs. time, $F_0$ is the background subtracted initial intensity value of the fluorescence. Background fluorescence was subtracted from each data point and was normalized using $F_0$ as a reference point. Finally, changes in average fluorescence intensity was graphically represented for each condition.

**Animals studies.** All the procedures were approved and carried out in accordance with the guiding ethical principles of the Institutional Review Board (SGH). Written informed consent was obtained for use of these human samples for the specific research purpose only. All procedures and protocols used in this study were approved by the Biological Resource Centre (BRC) Institutional Animal Care and Use Committee (#IACUC 150158 and 161111). Tumor samples used for generating the ovarian xenografts was established (Details in supplementary materials and methods). Clinical data of the patient-derived tumors are in Supplementary Table 3. MEFs were obtained from littermates p53$^{-/-}$, p53$^{+/+}$ and p53$^{+/-}$ mice, and p53$^{+/+}$, p53$^{+/R172H}$ and p53$^{R172H/R172H}$ mice[55].

## Data availability

All data needed for the interpretation of the results are presented in this paper; original source of big data sets are quoted in Methods section.

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

## Acknowledgements

We thank Bert Vogelstein for the HCT116 isogenic cell lines, Dr Graham Wright (Institute of Medical Biology Microscopy Unit) for help with live cell imaging and microscopy and all members of the CCF lab/IMCB. We thank Dr. Brian Chia, Experimental Therapeutics Centre, A*STAR, Singapore, for help with the chemistry and design of niclosamide analog. This work was supported by funding from IFOM, FIRC Institute of Molecular Oncology, A*STAR Agency for Science Technology and Research, Singapore and Polaris program grant (SPF2012/001) under the A*STAR Strategic Positioning Fund. We sincerely apologize to authors whose articles are not cited here due to space constraints.

## Author contributions

Conceptualization, C.F.C.; Methodology, R.K., L.C., Y.S.H., A.R., and M.L.; Validation, R. K., B.S., P.Y.H., O.A.; Formal Analysis, A.R., B.S., O.A., H.I., S.Y.T., L.C., H.Y., C.Y.L., and R.K.; Investigation, Y.S.H., P.Y.H., S.C., S.Y.M., H.I., H.Y., R.K., and L.C.; Resources, B.C.G., T.H.H., Y.K.L., S.H.C., and A.P.C.W.; Writing—Original Draft, C.F.C.; Writing—Review & Editing, L.C., R.K., C.Y.L., and C.F.C.; Visualization, R.K., L.C. and C.F.C.; Supervision and Funding Acquisition, C.F.C.

## Additional information

**Competing interests:** The authors declare no competing interests.

