## [Peer Review File · Nature Communications]

Reviewers' comments:

Reviewer #1 (Remarks to the Author):

The manuscript by Kumar et al describes the use of Niclosamine as a cancer therapeutic that preferentially kills p53 mutant/deficient cells. This is a very thorough and comprehensive study. Aside from a number of typographical/grammatical errors, the study is well described. I have no major concerns, but here are a few comments and points of clarification that I have:

1. Page 15- The authors use AA to attempt to prove that increase in AA "...induces apoptosis through a common mechanism..." As compared to Niclosamide. However inducing apoptosis does not necessarily mean that the mechanisms is the same. Does AA treatment induce mitochondrial uncoupling?

2. The use of doxorubicin in figure 6 seems to come out of nowhere. I assume that the authors are using it as a p53 inducer, however Doxo has pleiotropic effects and it would be nice to have a "cleaner" inducer of p53. Perhaps an inducible genetic p53 model or a more direct p53 inducer like Nutlin.

3. The authors claim that ALOX5 and ALOX 12B are novel direct targets of p53. This claim would be strengthened by showing that p53 directly binds to these promoters, perhaps with CHIP.

4. In the text all references to figure 7 are mis-labeled. They are off by one letter in reference to the actual figure panels.

Reviewer #2 (Remarks to the Author):

This is an interesting study demonstrating an induced vulnerability of p53 null cells to treatment with the drug niclosamide. The data support a model in which mitochondrial uncoupling induced by niclosamide leads to calcium release, which promotes generation of arachidonic acid. In p53 wild type cells, p53 dependent induction of the lipid oxygenation enzymes ALOX5 and ALOX12 further metabolise AA and limit toxicity of niclosamide treatment. Loss of p53 results in the accumulation of AA and cell death. Overall this is a very nice paper – there are a couple of points that should be addressed.

1. Using matched cell lines or a series of naturally occurring mutant and wtp53 cancer cells, the authors show convincing evidence that loss of p53 leads to a sensitivity to niclosamide (Figures 1 and 2). The authors then show that there is a change in mitochondria and cytochrome c release, but do not show directly that this is the cause of the cell death (as claimed on page 10). I agree it seems likely to be the case, but the authors may wish to soften their claim – at present they simply show a correlation. Could the authors speculate on why does induction of PUMA not kill these cells?

2. Central to the argument is the effect of niclosamide on mitochondrial uncoupling and calcium release. Do other mitochondrial uncouplers show the same effect?

3. I would like to see a more complete analysis of the changes in fatty acids that are identified in the MEFs in the HCT116 cells – not just arachidonic acid (Figure 4). The authors must have these data in their analyses.

4. Is AA known to induce cytochrome c release and apoptosis? Can the authors explain this response?

5. While there does seem to be p53-dependent expression of ALOX5 and ALOX12, the authors should comment on whether these two genes are direct transcriptional targets of p53. Do they contain p53-binding sites? Can some evidence for increased protein levels be provided?

5. In Figure 7, the authors need to show the response of p53 null cells to ALOX5 and ALOX12 knock down. This is an important control, although these cells are more sensitive to niclosamide, according to the model this sensitivity should not be affected by ALOX5 and ALOX12 depletion.

6. The in vivo experiment shown in Figure 8 is good, but I wonder why the authors suddenly move to this PDX system? HCT116 cells form xenograft tumours and the study would be more compelling if a similar in vivo response were shown using these isogenic lines. At present, it's hard to conclude that the different response of KKH02 and KKH011 is really due to p53 status.

Minor point: There is a problem with the labelling of Figures 5 and 7.

.

Reviewer #3 (Remarks to the Author):

In the manuscript entitled “Mitochondrial uncoupling reveals a novel therapeutic target for p53 defective cancers” Kumar Coronel et al. show that the repurposed drug niclosamide preferentially decrease the viability of p53 defective cells compared to p53 wild type cells. This finding is recapitulated in multiple cell models in vitro, and in a PDX model of ovarian cancer. The authors propose that the mechanism by which niclosamide preferentially affects p53 defective cells, involves mitochondrial metabolism, calcium signaling and arachidonic acid (AA) metabolism. The paper is well written and the scientific narrative is fluent. However it is unclear which novel therapeutic target is revealed by the study. The following points should be addressed before publication.

1. The mechanism proposed is vague and the results presented only suggest the reasons why p53 defective cells are more sensitive to the niclosamide. The authors hypothesize that the p53-dependent induction of ALOX5/ALOX12B, enzymes responsible for the breakdown of arachidonic acid, is key in increasing the sensitivity of p53 defective cells to niclosamide. If this hypothesis is correct niclosamide should increase the levels of AA in a p53-dependent way. Moreover, if ALOX5/ALOX12B are responsible for the increased sensitivity to niclosamide of p53-defective cells, the antiproliferative/cytotoxic effects of niclosamide should be equalized in p53 WT and mutant cells by ALOX5/ALOX12B silencing. The authors should perform experiments to address these points.

2. To prove that the difference in AA metabolism between p53-WT and p53-defective cells is responsible for the differential sensitivity to niclosamide in vivo, the xenografts should be treated with an inhibitor of the AA biosynthesis, such as cPLA2 α inhibitor, in combination with niclosamide. Indeed, according to the proposed working models, the inhibition of the AA biosynthesis should rescue the growth of niclosamide-treated p53 mutant xenografts.

3. The Authors should strengthen the in vivo results by testing the effects of niclosamide in more than one “cherry picked” PDX model. The number of different PDXs in which niclosamide will have been tested should be openly reported in the manuscript.

4. The AA levels should be measured in the xenograft models treated with niclosamide and compared to untreated.

5. The authors show in Sup Fig 2 that niclosamide uncouple mitochondrial respiration similarly to FCCP. According to the working model, the uncoupling effect of niclosamide affects calcium homeostasis, which in turn affect AA metabolism. If this model is correct, FCCP should phenocopy the effects of niclosamide on calcium and AA metabolism. On the contrary, the authors should revisit the causal role attributed to the uncoupling property of niclosamide.

6. The effects of niclosamide on calcium flux in p53 +/+ and -/- cells are different in kinetics but comparable in intensity (compare $\Delta F/F_0$ in Fig 5 a and b). How can a 60-minute delay in the maximal calcium flux response affect the steady state levels of AA, and in the long term change cell viability and tumor growth? The causal role of calcium "flux" in mediating the differential sensitivity to niclosamide should be tested pharmacologically by using calcium chelators, as well as inhibitors of calcium release from cellular organelles.

Reviewer #1 (Remarks to the Author):

The manuscript by Kumar et al describes the use of Niclosamine as a cancer therapeutic that preferentially kills p53 mutant/deficient cells. This is a very thorough and comprehensive study. Aside from a number of typographical/grammatical errors, the study is well described. I have no major concerns, but here are a few comments and points of clarification that I have:

Thank you Reviewer 1 for the positive comments. We have addressed Reviewer's 1 concerns as detailed below.

1. Page 15- The authors use AA to attempt to prove that increase in AA "...induces apoptosis through a common mechanism..." As compared to Niclosamide. However inducing apoptosis does not necessarily mean that the mechanisms is the same. Does AA treatment induce mitochondrial uncoupling?

Thank you to Reviewer 1 for pointing this out.
We have reworded the text to better reflect our suggestion.

What we had meant is that the downstream effects of AA in inducing cytochrome c release is common with niclosamide action. And indeed, we did not find that AA treatment induce mitochondrial uncoupling.

We have rephrase the text to (Page 15, line 14) : "AA and niclosamide induce apoptosis potentially through a common pathway involving cytochrome C release."

2. The use of doxorubicin in figure 6 seems to come out of nowhere. I assume that the authors are using it as a p53 inducer, however Doxo has pleiotropic effects and it would be nice to have a "cleaner" inducer of p53. Perhaps an inducible genetic p53 model or a more direct p53 inducer like Nutlin.

We agree with the reviewer's suggestion, and have performed experiments using Nutlin as a more direct inducer of p53 transcriptional activity. We demonstrated that Nutlin potently induced the gene expression of ALOX5 and ALOX12B in wildtype p53 cells, similar to p21 gene, but not in the isogenic p53KO cells (Fig. 6f). These results further strengthen the data and suggestion that these genes are bona-fide p53-induced target genes.

Text change are included Page 19 Line 1 "To test if these genes are also induced in response to a more specific inducer of p53 activity, we used an MDM2 antagonist, Nutlin (Vassilev et al 2004 Science), which blocks the binding of MDM2 to p53, resulting in p53 stabilization and activation. Indeed, both ALOX5 and ALOX12B, similar to p21, were potently induced by nutlin in wildtype HCT116 cells but not in the p53^{-/-} cells, as were expected of p53-induced genes (Fig. 6f)."

3. The authors claim that ALOX5 and ALOX 12B are novel direct targets of p53. This claim would be strengthened by showing that p53 directly binds to these promoters, perhaps with ChIP.

In line with the reviewer request, we have performed ChIP-qPCR experiments using a p53-specific antibody. We demonstrated in Supplementary Fig. S6f and S6g that p53 binding is enriched on the intragenic regions within the ALOX5 and ALOX12B genes.

The same intragenic regions were also found by bioinformatics analysis of established p53 ChIP-seq data ((GSM1142696(McDade et al., 2014)) (Supplementary Fig. S6f and S6g; upper panels). Our ChIP-qPCR data showed reproducibly that there is obvious significant p53 enrichment at the indicated p53 peaks, in contrast to other irrelevant regions that are devoid of p53 binding (as seen in analysed p53 ChIP-seq data; Supplementary Fig. S6f-g; upper panels), and therefore confirmed that p53 interacts with specific intragenic loci on ALOX5/12B genes. Accordingly, many reports have demonstrated that p53 can bind in introns, away from the promoter regions, that serve as enhancers to promote the transcriptional activation of genes.

Together these data strongly suggest that ALOX5 and ALOX12B are novel direct targets of p53, consistent with the data in Fig. 6b, 6c and 6f showing a transcriptional upregulation of these genes in a p53-dependent manner, in response to both nutlin and doxorubicin.

We included the text changes in Page 20, Line 2.

4. In the text all references to figure 7 are mis-labeled. They are off by one letter in reference to the actual figure panels.

Thank you to reviewer 1 for pointing this out. This has now been corrected in the text.

Reviewer #2 (Remarks to the Author):

This is an interesting study demonstrating an induced vulnerability of p53 null cells to treatment with the drug niclosamide. The data support a model in which mitochondrial uncoupling induced by niclosamide leads to calcium release, which promotes generation of arachidonic acid. In p53 wild type cells, p53 dependent induction of the lipid oxygenation enzymes ALOX5 and ALOX12 further metabolize AA and limit toxicity of niclosamide treatment. Loss of p53 results in the accumulation of AA and cell death. Overall this is a very nice paper – there are a couple of points that should be addressed.

Thank you to Reviewer 2 for the positive comments. We have addressed Reviewer 2 concerns as below

1. Using matched cell lines or a series of naturally occurring mutant and wtp53 cancer cells, the authors show convincing evidence that loss of p53 leads to a sensitivity to niclosamide (Figures 1 and 2). The authors then show that there is a change in mitochondria and cytochrome c release, but do not show directly that this the cause of the cell death (as claimed on page 10). I agree it seems likely to be the case, but the authors may wish to soften their claim – at present they simply show a correlation. Could the authors speculate on why does induction of PUMA not kill these cells?

Thanks for pointing this out. We have reworded the text to soften the claim We rephrased some words as stated below (the underlined words replaced the original text):

(Page 9 line 21) “We further demonstrated that p53-independent mechanism induced by niclosamide was correlated to mitochondrial dysfunction and cytochrome c release from

the mitochondria”

(Page 10 line 3) “The results are consistent with the suggestion that a programmed mitochondrial death pathway comprising of the reported apoptosome cytochrome/APAF1/Cas-9 may be activated in p53-deficient cells in response to niclosamide, potentially leading to an irreversible apoptotic signaling cascade targeting caspase-3 and PARP1”

(Page 15 line 14) “Together, our data suggested that AA and niclosamide potentially induce apoptosis through a common pathway involving cytochrome c release from the mitochondria...”

(Page 25 line 18) “Loss of wildtype p53 activity results in a marked accumulation of arachidonic acid that correlates to an increase in cytochrome c release and caspase9/3 activation.”

We did not find that the induction of PUMA by niclosamide-induced activation of p53 is sufficient to kill the wildtype cells. Indeed we observed that the activation of PUMA induced by a low dose of an inducing agent do not always lead to apoptosis (Cheek et al., 2007). The reason may be that the final decision for apoptosis is determined by a number of other factors, for example, the levels of anti-apoptotic Bcl-2 (that PUMA binds to) that controls the intrinsic apoptotic pathway. Therefore, the final cell fate (survival or apoptosis) may be controlled by a number of factors including, other than PUMA, other proteins in the Bcl-2 pathway (Shamas-Din et al., 2013).

2. Central to the argument is the effect of niclosamide on mitochondrial uncoupling and calcium release. Do other mitochondrial uncouplers show the same effect?

To address this, we demonstrated that a classical mitochondrial uncoupler FCCP also showed an increase in calcium signaling, similar to niclosamide, as measured in live cell calcium imaging using Fluo-4 dye (Supplementary Fig. 5e-h). We showed that FCCP also induce a rapid increase in intracellular calcium signals almost immediately, similar to the kinetics seen with niclosamide. In addition, we demonstrated that FCCP, similar to niclosamide, also sensitizes p53-deficient cells in colony and cell viability assays (Supplementary Fig S5c and S5d). Together, our data support the working model that central to the differential effects of niclosamide on WT and KO cells is its activity on mitochondrial uncoupling and calcium release.

We included the changes in the main text Page 13 Line 21 ““To further determine if mitochondrial uncoupling is an important biochemical function of niclosamide related to its effects in sensitizing p53-deficient cells, we tested another known mitochondrial uncoupler, FCCP, using another mitochondrial uncoupler, FCCP, which phenocopied the effects niclosamide on cellular sensitivity, calcium and AA metabolism.”

3. I would like to see a more complete analysis of the changed in fatty acids that are identified in the MEFs in the HCT116 cells – not just arachidonic acid (Figure 4). The authors must have these data in their analyses.

We have now included a graph of the change in metabolites identified in MEFs (Fig 4a) in HCT116 cells (Supplementary S4c). Of note, other than the key fatty acid of interest, arachidonic acid (AA 20:4), eicosapentaenoic acid (EPA 20:5) is also further increased in p53-/- when compared to the p53+/+ cells, similar to that observed in MEFs (Fig 4a).

The same is true of the LysoPEs (Fig S4c). A few fatty acids differentially upregulated in mutant MEFs (for example, FA 22:6 and FA22:1) were found to be similarly upregulated in HCT116 p53^{+/+} and p53^{-/-} cells, which could be due to the different cell-type dependent processing of these fatty acids that we currently do not understand. Nevertheless, the major fatty acid of interest, AA 20:4 is reproducibly enriched in p53-deficient cells of both human (HCT116) and mouse origins (MEFs) in response to niclosamide treatment, which is consistent with AA being an important effector of niclosamide response.

4. Is AA known to induce cytochrome c release and apoptosis? Can the authors explain this response?

Indeed, Scorrano L et al 2001 suggested that AA induces cytochrome c release and cell death through the induction of mitochondrial permeability transition (PT). It is known that mitochondrial PT is regulated by the opening of PT pore, that can eventually lead to the release of proapoptotic proteins such as cytochrome c. This is consistent with our findings that niclosamide induces AA release, which promotes cytochrome c release and apoptosis. More importantly, we demonstrated that this phenomenon is further exacerbated upon p53 loss, thus explaining the preferential sensitivity of p53-deficient cells towards niclosamide.

We made text changes to include this (Page 15 line 17)

5. While there does seem to be p53-dependent expression of ALOX5 and ALOX12, the authors should comment on whether these two genes are direct transcriptional targets of p53. Do they contain p53-binding sites? Can some evidence for increased protein levels be provided?

To address this, we performed p53 ChIP-qPCR experiments to verify if p53 binds to the genomic loci of ALOX5 and ALOX12B genes. We demonstrated that p53 is enriched at specific intragenic sites in ALOX5 and ALOX12B genes (Supplementary Fig. S6f and S6g). The qPCR analyses also confirmed that the detected p53 enrichment exist at the same intragenic sites shown by bioinformatics analysis of established p53 ChIP-seq data ((GSM1142696(McDade et al., 2014)) (Supplementary Fig. S6f and S6g; upper panels). Accordingly, many reports have demonstrated that p53 can bind in introns, away from the promoter regions, and these intronic p53 binding sites serve as enhancers for the transcriptional activation of genes. Together the data suggest that ALOX5 and ALOX12B are direct transcriptional targets of p53.

Text changes are included :
(Page 20 Line 1)

“Finally, to determine if these genes are direct p53 targets, we first examined the interaction of p53 with the genomic loci of ALOX5 and ALOX12B using bioinformatics analysis of established p53 ChIP-seq database (GSM1142696(McDade et al., 2014)). We visualized p53 binding peaks on *ALOX12B* and *ALOX5* in the UCSC Genome Browser using reference genome hg38. As shown, distinct p53 peaks were detected at ALOX5 and ALOX12B genomic loci (Supplementary Fig 6f and 6g; Upper panels). To further verify if these are p53 binding sites, we performed ChIP experiments. We designed primers to amplify genomic regions that overlap with the putative p53 peak or sites (A and B) that do not appear to bind p53 (Supplementary Fig 6e; Upper panel). We

incubated sonicated lysates prepared from HCT116 p53^{+/+} and p53^{-/-} cells with a p53-specific antibody (DO-1) and examined the binding of p53 to ALOX5 genomic loci using site-specific qPCR primers. qPCR analyses revealed that p53 specifically binds to the region corresponding to the putative p53 peak but not to the irrelevant regions A and B.

Similarly, we explored the interaction of p53 with the ALOX12B gene locus. ChIP-qPCR analyses revealed a specific enrichment of p53 binding to the indicated genomic regions, Peak I and II, but not to the irrelevant region C (Supplementary Fig 6g; Upper panel). Together, the data demonstrate that p53 binds specifically to ALOX5 and ALOX12B genomic loci within intragenic regions and suggested that ALOX5 and ALOX12B are direct gene targets of p53. The binding of p53 to intragenic regions is consistent with reports suggesting that p53 can bind to enhancer sites away from promoter regions to promote transcription activation (Smeenk et al., 2008; Wei et al., 2006). ”

In addition, we detected an increase in protein expression of ALOX5 and ALOX12B in HCT116 wildtype cells compared to p53KO cells, which is further enriched in response to a specific inducer of p53 activity Nutlin (Supplementary Fig. 6c).

5. In Figure 7, the authors need to show the response of p53 null cells to ALOX5 and ALOX12 knock down. This is an important control, although these cells are more sensitive to niclosamide, according to the model this sensitivity should not be affected by ALOX5 and ALOX12 depletion.

Thank you for the excellent suggestion. We have now completed the experiment including the p53^{-/-} cells harboring knockdown of ALOX5 and ALOX12B. As demonstrated in Supplementary Fig. S7a, knockdown of ALOX5 and ALOX12B sensitizes wildtype HCT116 cells to niclosamide. However, in HCT116 p53^{-/-} cells treated with niclosamide, little or no effects of ALOX5 and ALOX12B depletion were observed, consistent with the explanation that both genes are wildtype p53-dependent, and controls cellular sensitivity to niclosamide in the wildtype cells.

Text changes are included:

Page 22 Line 20,

“As expected, knockdown of *ALOX5* or *ALOX12B* barely further compromised the viability of p53^{-/-} cells in response to niclosamide (Supplementary Fig. S7a). Together, our data supports the model that *ALOX5* and *ALOX12B* are important gene targets upregulated by p53 activation in wildtype cells, which opposes niclosamide-induced cytotoxicity.”

6. The in vivo experiment shown in Figure 8 is good, but I wonder why the authors suddenly move to this PDX system? HCT116 cells form xenograft tumours and the study would be more compelling if a similar in vivo response were shown using these isogenic lines. At present, it's hard to conclude that the different response of KKH02 and KKH011 is really due to p53 status.

To address the reviewer's comment, we have performed an additional xenograft experiment using HCT116 isogenic lines.

As demonstrated in Fig 8a, oral gavage with niclosamide of NSG mice harboring HCT116 p53^{-/-} tumor xenografts with niclosamide led to a significant tumor growth delay; achieving a percentage tumor growth inhibition of 45.7%. However, HCT116 p53^{+/+}

xenografts showed a significantly reduced percentage of tumor growth inhibition of 15.6% instead (Fig 8b). These results are consistent with the increased efficacy of niclosamide in PDX model KKH02 (mutant p53) when compared to KKH011 (WT p53) (Fig 8e and 8f). Together our results supported the working model that the p53 loss of function mutations/deletion increases the *in vivo* efficacy of niclosamide on tumor inhibition.

We have included the text changes in Page 23 Line 11

“To illustrate the impact of the genetic status of p53 on the efficacy of niclosamide *in vivo*, we injected isogenic HCT116 p53^{+/+} and p53^{-/-} cells to grow as tumor xenografts in nude mice and performed oral dosing of niclosamide for 28 days. We observed a significant tumor growth delay in HCT116 p53^{-/-} tumors when mice are dosed with niclosamide; tumor growth inhibition measured at the end of study is 45.7% (Fig. 8a). In contrast, insignificant growth delay was observed in HCT116 p53^{+/+} tumors in the treated cohort, achieving only 15.7% tumor growth inhibition compared to control (Fig. 8b). Kaplan-Meier analysis of the tumor growth delay data, and comparison of the niclosamide-treated and vehicle treated groups in the p53KO tumor cohort using log-rank t test analysis showed that both curves are statistically significantly different (P<0.05), in contrast to the p53 wildtype tumor cohort which showed a insignificant difference between drug-treated and vehicle control groups (P>0.05) (Supplementary Fig. 8a and 8b). Consistently, we observed an increased in cleaved caspase 3 in HCT116 p53^{-/-} tumors sections (Supplementary Fig. 8c). ”

Minor point: There is a problem with the labelling of Figures 5 and 7.

Thank you for pointing this out, we have now corrected the labeling in the revised version.

Reviewer 3

In the manuscript entitled “Mitochondrial uncoupling reveals a novel therapeutic target for p53 defective cancers” Kumar Coronel et al. show that the repurposed drug niclosamide preferentially decrease the viability of p53 defective cells compared to p53 wild type cells. This finding is recapitulated in multiple cell models *in vitro*, and in a PDX model of ovarian cancer. The authors propose that the mechanism by which niclosamide preferentially affects p53 defective cells, involves mitochondrial metabolism, calcium signaling and arachidonic acid (AA) metabolism. The paper is well written and the scientific narrative is fluent. However it is unclear which novel therapeutic target is revealed by the study. The following points should be addressed before publication.

Thank you Reviewer 3 for the comments. We have addressed Reviewer’s 3 concerns as detailed below.

1.The mechanism proposed is vague and the results presented only suggest the reasons why p53 defective cells are more sensitive to the niclosamide. The authors hypothesize that the p53-dependent induction of ALOX5/ALOX12B, enzymes responsible for the breakdown of arachidonic acid, is key in increasing the sensitivity of p53 defective cells to niclosamide. If this hypothesis is correct niclosamide should increase the levels of AA in a p53-dependent way.

As the reviewer has rightly pointed out, niclosamide should affect the levels of AA dependent on the status of p53. Indeed, we show in Fig 4a, 4b and 4d that the levels of AA (20:4) measured using tandem LC-MS are higher in the p53 defective cells, compared to the wildtype cells. This is consistent with our model, exactly as the reviewer has also suggested, that wildtype p53-dependent induction of *ALOX5* and *ALOX12B* promotes the breakdown of arachidonic acid in wildtype p53 cells, while p53-defective cells accumulate arachidonic acid since levels of *ALOX5* and *ALOX12B* are lower in the absence of p53.

Moreover, if *ALOX5/ALOX12B* are responsible for the increased sensitivity to niclosamide of p53-defective cells, the antiproliferative/cytotoxic effects of niclosamide should be equalized in p53 WT and mutant cells by *ALOX5/ALOX12B* silencing. The authors should perform experiments to address these points.

This is an excellent suggestion by the reviewer. We have performed experiments by knocking down *ALOX5* and *ALOX12B* in isogenic HCT116 p53^{+/+} and p53^{-/-} cells. We had demonstrated that silencing of *ALOX5/12B* sensitizes wildtype HCT116 cells to niclosamide (Fig 7a-c). Now we showed that while depletion of *ALOX5* and *ALOX12B* strongly compromised cell viability in HCT116 p53^{+/+} cells, *ALOX5/ALOX12B* depletion did not further exacerbate the response of p53^{-/-} cells to niclosamide, as expected (Supplementary Fig 7a). More importantly, depleting *ALOX5* and *ALOX12B* in p53^{+/+} cells resulted in loss of cell viability in response to niclosamide at similar doses (2-3 μ M) of the drug that was effective in p53^{-/-} cells (Supplementary Fig 7a).

We included the text changes Page 22 Line 20,

“As expected, knockdown of *ALOX5* or *ALOX12B* in p53^{-/-} cells barely further compromised the viability of p53^{-/-} cells in response to niclosamide (Supplementary Fig S7a). This result supports the model that *ALOX5* and *ALOX12B* are important gene targets upregulated by p53 activation in wildtype cells, which prevented niclosamide-induced cytotoxicity.”

2.To prove that the difference in AA metabolism between p53-WT and p53-defective cells is responsible for the differential sensitivity to niclosamide in vivo, the xenografts should be treated with an inhibitor of the AA biosynthesis, such as cPLA2 α inhibitor, in combination with niclosamide. Indeed, according to the proposed working models, the inhibition of the AA biosynthesis should rescue the growth of niclosamide-treated p53 mutant xenografts.

We agree with the reviewer that experiments using inhibitors of the AA biosynthesis pathway would further strengthen the link we have made between AA accumulation and niclosamide-induced cytotoxicity. We have now tested the effects of two cPLA2 α inhibitors on niclosamide-induced cytochrome c release, and showed that addition of a cPLA2 α inhibitor clearly suppressed the robust release of cytochrome c in niclosamide-treated p53-deficient cells (Fig 5g) and also partially rescued the growth of niclosamide-treated p53-deficient cells (Fig 5h). This is in line with the suggestion that the pronounced AA accumulation in niclosamide-treated p53-deficient cells is responsible for the differential sensitivity to niclosamide.

Text changes are included : (Page 16 line 1)

“To further illustrate that the effects of niclosamide is mediated through AA release, we utilized chemical inhibitors of calcium-dependent cytosolic phospholipase A2 (cPLA2). We selected two specific inhibitors of cPLA2 and tested if they would attenuate niclosamide-induced release of cytochrome c. Indeed, we found that inhibition of cPLA2 suppressed the extent of cytosolic cytochrome c induction by niclosamide upon p53 loss (Fig. 5g). Furthermore, inhibiting cPLA2 also rescued partially the colony growth of cells treated with niclosamide (Fig. 5h). The PLA2 inhibitors on their own did not significantly or adversely affect cell viability at the concentrations tested (Supplementary Fig. S5j). Together, our results concur with the suggestion that niclosamide induces cytochrome c release and apoptosis, at least in part, through AA release.”

We did not, however, carry out similar tests *in vivo*, as we believe that pleiotropic systemic effects (eg. changes in inflammatory responses and vasopermeability) induced upon blocking PLA2 signaling as reported (Kim et al., 2016; Kisslov et al., 2012; Linkous et al., 2009; Patel et al., 2008), will likely obscure any direct response on AA biosynthesis in the xenografts and prevent meaningful conclusions to be made between AA accumulation and niclosamide-induced reduction of tumor proliferation *in vivo*.

Nevertheless, the mechanistic interpretation of our data using cPLA2 α inhibitors (Fig 5g and 5h), which suppress AA biosynthesis, allowed us to strongly suggest that the cytotoxicity induced by niclosamide is dependent on AA accumulation.

Furthermore, we also demonstrated that inhibitors of calcium signaling clearly attenuate niclosamide effects, in line with the suggestion that activation of cPLA2 by calcium signaling is needed for niclosamide-induced cytotoxicity (point 6 below).

3. The Authors should strengthen the *in vivo* results by testing the effects of niclosamide in more than one “cherry picked” PDX model. The number of different PDXs in which niclosamide will have been tested should be openly reported in the manuscript.

We apologize that the reviewer appears to be under the misconception that we have tested multiple PDXs but selectively presented (‘cherry-picked’) the two that best fit our model. We would like to clarify that we have only tested KKH011 and KKH02 PDXs for response to niclosamide in the current study. KKH011 and KKH02 were paired for p53 wildtype and mutant genotypes and were used for the test on niclosamide response because the tumors exhibited comparable growth kinetics *in vivo*, a critical criterion for *in vivo* tumor growth/inhibition assays. While more pairs of PDXs would doubtlessly strengthen our conclusions, we were unfortunately constrained by availability of additional suitable PDXs within the current scope of this study. We have now modified our text to indicate this clearly.

Though we were unable to carry out additional studies using PDXs, we have further strengthened the *in vivo* results by testing xenografts of isogenic cell lines harboring WT p53 or knockout at the p53 gene locus, which is well validated for *in vivo* studies as reported (Maddocks et al., 2013; Ravi et al., 2000; Zhang et al., 2012). We showed in Fig 8a-b that HCT116 p53^{-/-} xenograft tumors respond better to niclosamide than HCT116 p53^{+/+} tumors, as demonstrated by the increased percentage of tumor growth inhibition (45.7% vs 15.6%). This is congruous with the patient derived xenograft (PDX) tumor models of KKH02(mutp53) and KKH011(WTp53) in Fig 8e-f showing that niclosamide improved the growth reduction of p53-deficient tumors. The key advantage

of using the isogenic cell line tumor model is that both tumors in comparison (HCT116 p53^{-/-} and p53^{+/+}) are of similar genetic background, with the only difference being p53 genetic status. From the mechanistic point of view, these results strongly support the suggestion that deficiency in p53 increases the *in vivo* efficacy of niclosamide on tumor inhibition.

We have included the text changes in Page 23 Line 11

“To illustrate the impact of the genetic status of p53 on the efficacy of niclosamide *in vivo*, we injected isogenic HCT116 p53^{+/+} and p53^{-/-} cells to grow as tumor xenografts in nude mice and performed oral dosing of niclosamide for 28 days. We observed a significant tumor growth delay in HCT116 p53^{-/-} tumors when mice are dosed with niclosamide; tumor growth inhibition measured at the end of study is 45.7% (Fig. 8a). In contrast, insignificant growth delay was observed in HCT116 p53^{+/+} tumors in the treated cohort, achieving only 15.7% tumor growth inhibition compared to control (Fig. 8b). Kaplan-Meier analysis of the tumor growth delay data, and comparison of the niclosamide-treated and vehicle treated groups in the p53KO tumor cohort using log-rank t test analysis showed that both curves are statistically significantly different (P<0.05), in contrast to the p53 wildtype tumor cohort which showed a insignificant difference between drug-treated and vehicle control groups (P>0.05) (Supplementary Fig. 8a and 8b). Consistently, we observed an increased in cleaved caspase 3 in HCT116 p53^{-/-} tumors sections (Supplementary Fig. 8c). ”

4.The AA levels should be measured in the xenograft models treated with niclosamide and compared to untreated.

Following the reviewer’s request, we profiled the AA levels in treated and untreated HCT116 p53^{-/-} and p53^{+/+} tumors harvested at the end-point of the 28-day treatment regime. We found no significant difference in AA levels in treated vs untreated tumors analyzed using LC-MS.

This result was not completely unexpected due to several reasons. Firstly, since AA accumulation is linked to niclosamide-induced cell death, we reasoned that tumor cells that responded well to the drug with high AA buildup would have been lost by the endpoint of the treatment. Conversely, cells that survived the 28-day niclosamide treatment probably had the poorest response to the drug and did not accumulate AA sufficiently to induce cell death. Secondly, tumors are much more heterogenous compared to *in vitro* cultures, as they become vascularized by p53^{+/+} cells coming from the host animals. The presence of host-derived wild-type p53 cells, which do not accumulate AA in response to niclosamide, likely further confounded the analysis of AA accumulation in treated and untreated tumors. As unambiguous conclusions could not be drawn from this experiment, the results were not included in the revised manuscript.

However, to improve our manuscript and provide additional evidence that p53^{-/-} cells differentially accumulate AA upon drug-induced mitochondrial uncoupling, we have tested the effects of another mitochondrial uncoupler, FCCP, on AA metabolism, as the reviewer suggested below (point 5). We showed that FCCP also induced AA accumulation in p53^{-/-} cells to a greater extent compared to p53^{+/+} cells, preceding cell death (Fig 4 and Supplementary Fig 4c and S5), thus providing a stronger link between mitochondrial uncoupling with AA accumulation in the absence of p53 function.

5. The authors show in Sup Fig 2 that niclosamide uncouple mitochondrial respiration similarly to FCCP. According to the working model, the uncoupling effect of niclosamide affects calcium homeostasis, which in turn affect AA metabolism. If this model is correct, FCCP should phenocopy the effects of niclosamide on calcium and AA metabolism. On the contrary, the authors should revisit the causal role attributed to the uncoupling property of niclosamide.

According to the reviewer's excellent suggestion, we tested the effects of the classical mitochondrial uncoupler FCCP on calcium and AA metabolism.

We demonstrated that FCCP led to a robust increase in calcium signaling, similar to niclosamide, as measured in live cell calcium imaging using Fluo-4 dye (Supplementary Fig. 5e-h). In response to FCCP, a rapid induction of intracellular calcium signals was observed, similar to the kinetics seen with niclosamide, consistent with the suggestion that mitochondrial uncoupling can induce calcium signaling. We also showed by metabolome analysis of FCCP-treated p53WT and p53KO cells that FCCP induced cellular accumulation of AA to a greater extent in p53KO cells compared to p53WT cells (Supplementary Fig 5i). Additionally, we demonstrated that FCCP, similar to niclosamide, also sensitizes p53-deficient cells in colony and cell viability assays (Supplementary Fig S5c and S5d). Therefore, FCCP treatment indeed phenocopies the effects of niclosamide. The new data strongly support our working model that the mitochondria uncoupling and the consequential changes in calcium and AA metabolism, is central to the differential niclosamide-induced phenotypes observed in p53 WT and KO cells.

Major addition to text : Page 13 line 21

"To further determine if mitochondrial uncoupling is an important biochemical function of niclosamide related to its effects in sensitizing p53-deficient cells, we tested another known mitochondrial uncoupler, FCCP. We demonstrated that FCCP also sensitizes HCT116 p53^{-/-} cells to a greater extent than compared to the isogenic HCT116 p53^{+/+} cells, in both colony growth and cell viability assays (Supplementary S5c, S5d). Indeed, FCCP also acts to induce intracellular calcium flux in both HCT116 p53^{+/+} and p53^{-/-} cells when tested in live cell imaging using Fluo-4 AM dye (Supplementary Fig. S5e-h). Similar to niclosamide (Fig. 5a, 5b), we observed a rapid induction of intracellular calcium signal, within minutes after the addition of FCCP, and to a comparable extent in HCT116 p53^{+/+} and p53^{-/-} cells (Supplementary Fig. S5e and S5f). To further characterise the functional links between mitochondrial uncoupling, calcium flux and arachidonic acid accumulation observed in p53^{-/-} cells upon niclosamide addition, we performed an unbiased metabolome analysis in HCT116 p53^{+/+} and p53^{-/-} cells exposed to FCCP. Consistent with our model and prediction, we detected significantly greater fold enrichment in arachidonic acid in HCT116 p53^{-/-} cells than in p53^{+/+} cells post-treatment with FCCP (Supplementary Fig. S5i), akin to that observed with niclosamide treatment. Taken together, our results is consistent with the model that the mitochondrial uncoupling effects of niclosamide affects calcium homeostasis which in turn alters AA metabolism, and potentially explain the hitherto unreported link between niclosamide and the release of arachidonic acid from phospholipids. This was further illustrated using a different mitochondrial uncoupler, FCCP, which phenocopied the effects niclosamide on cellular sensitivity, calcium and AA metabolism."

6. The effects of niclosamide on calcium flux in p53 +/+ and -/- cells are different in kinetics but comparable in intensity (compare deltaF/F0 in Fig 5 a and b). How can a 60-

minutes delay in the maximal calcium flux response affect the steady state levels of AA, and in the long term change cell viability and tumor growth? The causal role of calcium “flux” in mediating the differential sensitivity to niclosamide should be tested pharmacologically by using calcium chelators, as well as inhibitors of calcium release from cellular organelles.

We apologize that the original text was insufficiently clear on the effects of niclosamide-induced calcium flux and its related implications for our proposed model.

We demonstrated that niclosamide was able to induce calcium “flux” in both wildtype and p53-defective cells. And as the reviewer rightly pointed out, the calcium fluxes induced in p53^{+/+} and p53^{-/-} cells were of comparable intensity ($\Delta F/F_0$ in Fig 5a and b), albeit with a 60sec delay in the maximal calcium flux response in the p53^{-/-} cells. We agree with the reviewer that a 60 sec delay in maximal calcium flux response is unlikely to affect the steady state levels of AA and change long-term cell viability in p53^{-/-} cells.

Instead, what we propose in our model (Fig 8g) is that niclosamide causes mitochondrial uncoupling, which leads to a calcium flux in both wild-type and p53-defective cells. The niclosamide-induced calcium flux in turn triggers an increase in AA biosynthesis in both cell types. However, as p53WT cells are able to activate expression of *ALOX5* and *ALOX12B*, AA is effectively catabolized and steady state levels of AA remain low in these cells. By contrast, p53^{-/-} cells fail to induce *ALOX5/12B*, resulting in accumulation of AA levels and apoptosis.

We now show further support of this model by demonstrating that silencing of *ALOX5* and *ALOX12B* genes equalize the effects of niclosamide in p53^{+/+} and p53^{-/-} cells (Supplementary Fig 7a), as suggested by this reviewer in point 1.

In order to clarify the above points, we have made some text changes Page 14, Line 21 “Furthermore, our data suggest that whereas the initial drug-induced calcium flux is independent of p53 status, however, the steady state levels of AA is clearly dependent on p53 status, raising the possibility that the turnover of AA is affected in p53-deficient cells.”

We have now further shown that the calcium flux induced upon niclosamide treatment is an important part of the mechanism leading to niclosamide-induced apoptosis in p53^{-/-} cells. 2 new experiments were carried out, following the reviewer’s suggestions. We have now demonstrated that preloading cells with BAPTA, a calcium chelator, significantly reduced niclosamide-induced calcium response and further attenuated niclosamide-induced apoptosis, as determined by the extent of PARP1 cleavage (Supplementary Fig. S5l). Furthermore, combining BAPTA and EGTA led to a further reduction in niclosamide-induced calcium response and remarkably reduced niclosamide-induced apoptosis (Supplementary Fig. S5m). We also showed that carbacyclin, which attenuate cellular calcium levels by inhibiting calcium release from intracellular stores mediated through IP3 signaling (Li et al., 1997; Tertyshnikova and Fein, 1998), was able to antagonize niclosamide-induced growth inhibition and restored partially the growth of cells challenged with niclosamide (Supplementary Fig. 5n). Together, these data supported the suggestion that rise in calcium levels are needed for niclosamide-induced apoptosis.

Text changes are included: Page 16 Line 13

“In line with the above suggestion that niclosamide may function through calcium-dependent cPLA2 activation and AA release, partly through calcium signaling, we tested

how a commonly used intracellular chelator of calcium ions, BAPTA, may affect niclosamide effects. We pre-loaded cells with BAPTA-AM, and demonstrated that BAPTA-AM significantly reduced niclosamide-induced calcium response, monitored using a Fluo-4-AM indicator (Supplementary Fig. S5k(i)). Combination with another calcium chelator, EGTA, further reduced calcium levels upon niclosamide treatment (Supplementary Fig. S5k(ii)). Next, we tested if calcium chelators will attenuate niclosamide-induced apoptosis, as indicated by the extent of PARP1 cleavage. Indeed, we found that BAPTA-AM significantly reduced the extent of PARP1 cleavage induced by various concentrations of niclosamide in p53-deficient cells (Supplementary Fig. S5l). Furthermore, the percentages of Annexin V positive apoptotic cells following niclosamide treatment were also significantly reduced by BAPTA-AM and EGTA addition (Supplementary Fig. S5m). Together, our data concur with the suggestion that rise in calcium levels plays a role in mediating the observed cytotoxic effects of niclosamide, at least in part.”

Hence, taken together, our data strongly suggest that calcium signaling plays a critical role in initiating changes in AA biosynthesis which subsequently led to the accumulation of AA in p53-deficient cells (due to a deficiency in ALOX5 and ALOX12B enzymes), promoting apoptosis (Model depicted in Fig 8g).

References

- Cheok, C. F., Dey, A., and Lane, D. P. (2007). Cyclin-dependent kinase inhibitors sensitize tumor cells to nutlin-induced apoptosis: a potent drug combination. *Molecular cancer research : MCR* 5, 1133-1145.
- Kim, E., Tunset, H. M., Cebulla, J., Vettukattil, R., Helgesen, H., Feuerherm, A. J., Engebraten, O., Maelandsmo, G. M., Johansen, B., and Moestue, S. A. (2016). Anti-vascular effects of the cytosolic phospholipase A2 inhibitor AVX235 in a patient-derived basal-like breast cancer model. *BMC cancer* 16, 191.
- Kisslov, L., Hadad, N., Rosengraten, M., and Levy, R. (2012). HT-29 human colon cancer cell proliferation is regulated by cytosolic phospholipase A(2)alpha dependent PGE(2) via both PKA and PKB pathways. *Biochimica et biophysica acta* 1821, 1224-1234.
- Li, Z., Lee, H. C., Bielefeldt, K., Chapleau, M. W., and Abboud, F. M. (1997). The prostacyclin analogue carbacyclin inhibits Ca(2+)-activated K⁺ current in aortic baroreceptor neurones of rats. *The Journal of physiology* 501 (Pt 2), 275-287.
- Linkous, A., Geng, L., Lyshchik, A., Hallahan, D. E., and Yazlovitskaya, E. M. (2009). Cytosolic phospholipase A2: targeting cancer through the tumor vasculature. *Clinical cancer research : an official journal of the American Association for Cancer Research* 15, 1635-1644.
- Maddocks, O. D., Berkers, C. R., Mason, S. M., Zheng, L., Blyth, K., Gottlieb, E., and Vousden, K. H. (2013). Serine starvation induces stress and p53-dependent metabolic remodelling in cancer cells. *Nature* 493, 542-546.
- McDade, S. S., Patel, D., Moran, M., Campbell, J., Fenwick, K., Kozarewa, I., Orr, N. J., Lord, C. J., Ashworth, A. A., and McCance, D. J. (2014). Genome-wide characterization reveals complex interplay between TP53 and TP63 in response to genotoxic stress. *Nucleic acids research* 42, 6270-6285.

Patel, M. I., Singh, J., Niknami, M., Kurek, C., Yao, M., Lu, S., Maclean, F., King, N. J., Gelb, M. H., Scott, K. F., *et al.* (2008). Cytosolic phospholipase A2-alpha: a potential therapeutic target for prostate cancer. *Clinical cancer research : an official journal of the American Association for Cancer Research* 14, 8070-8079.

Ravi, R., Mookerjee, B., Bhujwala, Z. M., Sutter, C. H., Artemov, D., Zeng, Q., Dillehay, L. E., Madan, A., Semenza, G. L., and Bedi, A. (2000). Regulation of tumor angiogenesis by p53-induced degradation of hypoxia-inducible factor 1alpha. *Genes & development* 14, 34-44.

Shamas-Din, A., Kale, J., Leber, B., and Andrews, D. W. (2013). Mechanisms of action of Bcl-2 family proteins. *Cold Spring Harbor perspectives in biology* 5, a008714.

Smeenk, L., van Heeringen, S. J., Koepfel, M., van Driel, M. A., Bartels, S. J., Akkers, R. C., Denissov, S., Stunnenberg, H. G., and Lohrum, M. (2008). Characterization of genome-wide p53-binding sites upon stress response. *Nucleic acids research* 36, 3639-3654.

Tertyshnikova, S., and Fein, A. (1998). Inhibition of inositol 1,4,5-trisphosphate-induced Ca²⁺ release by cAMP-dependent protein kinase in a living cell. *Proceedings of the National Academy of Sciences of the United States of America* 95, 1613-1617.

Wei, C. L., Wu, Q., Vega, V. B., Chiu, K. P., Ng, P., Zhang, T., Shahab, A., Yong, H. C., Fu, Y., Weng, Z., *et al.* (2006). A global map of p53 transcription-factor binding sites in the human genome. *Cell* 124, 207-219.

Zhang, Q., Zeng, S. X., Zhang, Y., Zhang, Y., Ding, D., Ye, Q., Meroueh, S. O., and Lu, H. (2012). A small molecule Inauhzin inhibits SIRT1 activity and suppresses tumour growth through activation of p53. *EMBO molecular medicine* 4, 298-312.

-

REVIEWERS' COMMENTS:

Reviewer #1 (Remarks to the Author):

The authors have comprehensively and impressively addressed all of my concerns. I think this is a much improved piece of work because of all the suggestions of the reviewers and the work performed by the authors.

Reviewer #2 (Remarks to the Author):

The authors have addressed all the comments to my satisfaction. I am happy to support publication of the revised document.

Reviewer #3 (Remarks to the Author):

The Authors extensively revised the manuscript and the study has been strengthened by the new results and relevant controls.

I still have a couple of points that should be addressed by the Authors before publication:

1) The title of the study is "Mitochondrial uncoupling reveals a novel therapeutic target for p53 defective cancers". As mentioned in my previous comments it is unclear which novel therapeutic target is revealed. Can the Authors explicitly name the novel therapeutic target? Is the target the mitochondrial electron transport chain? If so, the title is circular (i.e. Mitochondrial uncoupling reveals...ETC). To my understanding the molecular therapeutic target has not been identified, and perhaps the authors should use the terms "therapeutic opportunity" or "therapeutic agent" instead?

A more appropriate and precise title should be found for the study.

2) Concerning the lack of evidences that niclosamide actually modulates AA levels in tumours (response to point 4 of this reviewer). I agree that there are a number of possible explanations for this negative result. However, reasonable speculations cannot exclude neither that the drug acts through a different mechanism of action when used in vivo, nor that it has off-target effects selectively affecting p53mut tumours. This negative result should not hinder the publication of the study, however it must be shown in the main figures, alongside the positive evidences supporting the proposed model, and it should be overtly discussed in the main text (results/discussion).

Other points

- There is no tumour regression showed in Figure 8. What it is shown is a delay in tumour growth. I am sure that a cancer patient would appreciate the difference in terminology. Modify the text accordingly (page 24 line 20).

Point to point response to reviewers' comments

Reviewer #1 (Remarks to the Author):

The authors have comprehensively and impressively addressed all of my concerns. I think this is a much improved piece of work because of all the suggestions of the reviewers and the work performed by the authors.

Reviewer #2 (Remarks to the Author):

The authors have addressed all the comments to my satisfaction. I am happy to support publication of the revised document.

Reviewer #3 (Remarks to the Author):

The Authors extensively revised the manuscript and the study has been strengthened by the new results and relevant controls.

I still have a couple of points that should be addressed by the Authors before publication:

1) The title of the study is "Mitochondrial uncoupling reveals a novel therapeutic target for p53 defective cancers". As mentioned in my previous comments it is unclear which novel therapeutic target is revealed. Can the Authors explicitly name the novel therapeutic target? Is the target the mitochondrial electron transport chain? If so, the title is circular (i.e. Mitochondrial uncoupling reveals...ETC). To my understanding the molecular therapeutic target has not been identified, and perhaps the authors should use the terms "therapeutic opportunity" or "therapeutic agent" instead? A more appropriate and precise title should be found for the study.

We agree with the reviewer and have now edited the title to "therapeutic opportunity"

2) Concerning the lack of evidences that niclosamide actually modulates AA levels in tumours (response to point 4 of this reviewer). I agree that there are a number of possible explanations for this negative result. However, reasonable speculations cannot exclude neither that the drug acts through a different mechanism of action when used *in vivo*, nor that it has off-target effects selectively affecting p53mut tumours. This negative result should not hinder the publication of the study, however it must be shown in the main figures, alongside the positive evidences supporting the proposed model, and it should be overtly discussed in the main text (results/discussion).

We have displayed the data on the arachidonic acid level in xenograft tumors in Fig 8c, alongside the tumor growth delay graphs. The implication of this result is also elaborated in results section Page 18 Line 23.

"To test if the AA levels in *in vivo* HCT116 tumors reflects that observed in HCT116 cells *in vitro*, we harvested the xenografts at the end of the study for

LC-MS metabolome detection of AA. However, as shown in Fig. 8c, we detected almost equivalent levels of AA in HCT116 p53^{+/+} and p53^{-/-} tumors, in contrast to what we observed in cells (Fig. 4d). One plausible explanation is that since AA accumulation is linked to niclosamide-induced cell death, we reasoned that those cells with high AA accumulation would have been eliminated at the end of the xenograft study. The remaining tumor cells would probably have the poorest response to niclosamide and therefore did not accumulate sufficient AA to induce cell death. Additionally, tumors are much more heterogenous in nature compared to cell cultures, with a strong likelihood of host cells (wildtype for p53) vascularizing the xenografts. The presence of host-derived cells that are wildtype for p53 would cause additional complexities in accurately comparing the AA levels.”

Other points

- There is no tumour regression showed in Figure 8. What it is shown is a delay in tumour growth. I am sure that a cancer patient would appreciate the difference in terminology. Modify the text accordingly (page 24 line 20).

We agree with the reviewer and have now change tumor regression to “tumor growth delay” in the results section (Figure 8)